# A Paleolimnological Perspective on Arctic Mountain Lake Pollution

Vladimir Dauvalter [1,*], Zakhar Slukovskii [1], Dmitry Denisov [1] and Alina Guzeva [1,2]

1    Institute of the North Industrial Ecology Problems of Kola Science Center of RAS, 184209 Apatity, Russia
2    Institute of Limnology of St. Petersburg Federal Research Center of RAS,196105 Saint Petersburg, Russia
*    Correspondence: vladimir@dauvalter.com; Tel.: +7-921-287-1580

**Abstract:** The chemical composition of sediments from the Arctic mountain Lake Bolshoy Vudjavr, situated in the western part of the Russian Arctic zone, was studied. The lake has been under intense anthropogenic load for more than 90 years since the development of the richest apatite–nepheline deposits in the world started. A 27 cm thick sediment core was sampled in the central part of the lake at the maximum depth of 37.4 m. The concentrations of more than 50 elements were analyzed by the mass spectral method, ICP-MS. The lake sedimentation rate established from the change in the content of the radioactive isotope $^{210}$Pb was 2.3 mm/yr. The effluent from apatite–nepheline production and atmospheric fallout enrich the sediments of Lake Bolshoy Vudjavr with alkali and alkaline earth metals, N, P, Mn, Fe, Al compounds, rare earth elements, and trace elements (Sb, Cu, Zn, Pb, Bi, Nb, Ta, Th). Analysis of the forms of elements in the lake sediments showed that the studied elements are mainly found in stable fractions—mineral, acid-soluble, and associated with organic matter. The pollution of the sediments of Lake Bolshoy Vudjavr was assessed by the integral index PLI (Pollution Load Index) and CF (contamination factor). The PLI value sharply increased after the "Apatite" Plant had been launched and a large amount of wastewater from the mines had been released into the lake. The highest PLI values were detected in the sediment layers accumulated during the period 1990s–2000s. Sb (18.2), P (10.3), Sr (7.8), and La (6.0) have the maximum CF values among all the studied elements.

**Keywords:** arctic mountain lake; sediment pollution; mining effluents; trace elements

## 1. Introduction

The negative impact of human activities, including the extraction of minerals, on the environment has been known since the Iron Age and the Roman Empire [1–3]. Evidence of anthropogenic metal pollution associated with mining and metallurgical activities dates back to at least 5000 years ago [4–6]. The pollution level rose during ancient Greek and Roman times [7] as a consequence of the increased mining, smelting, and trading activities [8,9]. Metal production technologies during this period were simple and highly polluting, releasing high concentrations of metals into the atmosphere, hydrosphere, sediments, and soils [10–12]. The 20th century saw the peaks of metal pollution in many European countries, due to heavy metal production after World War II, rebuilding and development of cities and industry, and maximum consumption of leaded gasoline around the 1970s [13].

During the 20th century, the Russian Arctic, and Murmansk region in particular, experienced rapid industrialization [14]. The mining industry has been and remains a key factor in the economic development of Murmansk region with numerous mines, processing plants and tailings that pollute soils, rivers, and lakes [15]. The development of the world's richest apatite–nepheline deposits of the Khibiny alkaline massif [14] has led to the formation of runoff and emissions of the main rock-forming and trace elements and their subsequent deposition in the previously intact mountain environment [16–19]. Mining, crushing, ore dressing, and

tailings discharges are the largest sources of environmental pollution [15,16,20–22]. The input of elements in elevated contents relative to natural background contents adversely affects the water quality in the lakes surrounding regional mining centers disrupt the biogeochemical cycles and production processes, which in turn decreases the species diversity of aquatic organisms [15,18,19,21,23,24]. Elevated levels of elements accumulated in the sediments of reservoirs are a sensitive indicator of human activity [15,16,18,22]. Due to the low ability of reservoirs to self-purify [15,18,19,25], the anthropogenicimpact irreversibly changes the natural environment in the conditions of the Arctic: the transformed chemical composition of water and sediments of the reservoirs alters the complexes and groups of hydrobionts living in them [15,18,21,25]. Arctic lakes can provide a long-term perspective on environmental change, including long-range atmospheric transport [25–27] and pollutant deposition trends derived from sediment core studies [15–17,28]. Studying elemental deposition in the Khibiny lakes is critically important in the light of current pollution and climate change trends reported from other European mountain ecosystems [29]. However, to date, only a limited number of studies have been devoted to the change in the elements accumulation in recent centuries in the lakes of the Russian Arctic and the Khibiny Mountains in particular (for example, [30,31]).

Several studies around the world investigated trace element contamination of water, sediments, soil, and plants as a result of mining activities [15–17,19,22,25–27,32–34]. These studies show that the levels of element contamination around mines depend on the geochemical characteristics, history, and pace of mining of mineral deposits [35–41]. High pollution levels of terrestrial and aquatic ecosystems persist for many years after the cessation of mining activities, especially if no measures are taken to restore the environment [42–48].

Trace elements, such as Pb, Sr, rare earth elements (REE), preserved in lacustrine sediments allow for a historical reconstruction of element accumulation in the Khibiny lakes [15,17–19,49]. This method offers a retrospective approach in the absence of long-term monitoring data [26,27,31,50]. Although, the Khibiny have a century-long history of pre-industrial [18,19], industrial [14],and modern anthropogenic activity [15,17], the lake sediments in the Khibiny were not studied with the determination of the sedimentation rate and a large list of analyzed elements.

This study of sediments in Lake Bolshoy Vudjavr focuses on the industrial era. In order to reconstruct the history of pollution and assess the intensity of anthropogenic impact on the largest Khibiny lake, the input of a large list of elements into the lake located in close proximity to the mines and receiving their effluents was analyzed. The studied sediments accumulate major rock-forming and trace elements associated with mining activities. Further studies of element accumulation in mountain lakes are required to determine the extent of environmental transformation in the Khibiny massif.

## 2. Materials and Methods

### 2.1. Research Area

The Khibiny Mountains are the largest mountain range in Murmansk region. They are located in the center of the region and their area is 1300 km$^2$ (Figure 1) [15,51,52]. The tops of the mountains are plateau-like, the slopes are steep with separate snowfields. The highest point of the Khibiny is Mount Judychvumchorr (1200.6 m above sea level). The Kukisvumchorr and Chasnachorr plateaus are located in the center of the Khibiny. Cities of Apatity and Kirovsk are at the foot of the Khibiny.

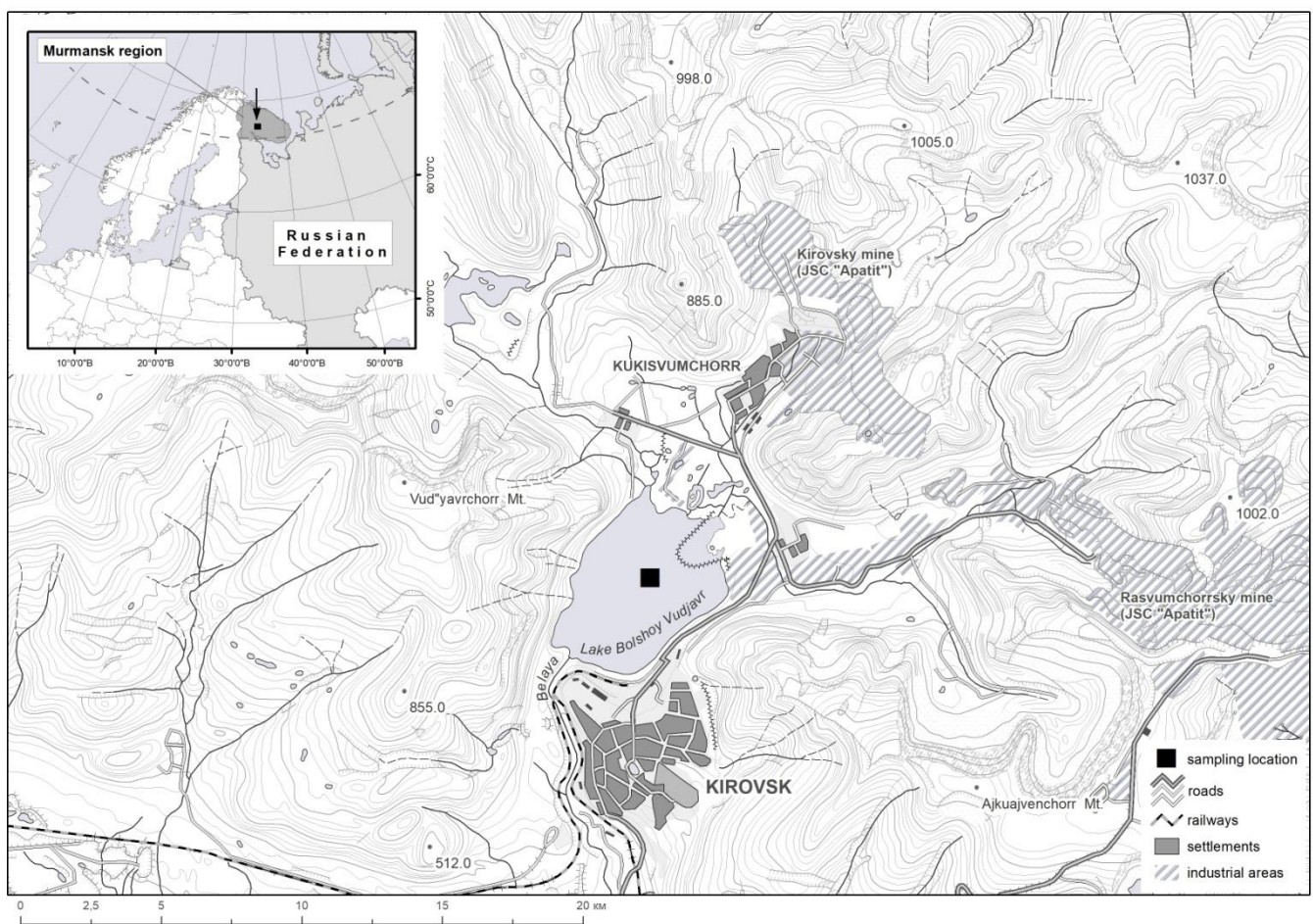

**Figure 1.** Map of the study area.

The climatic conditions in the Khibiny are harsh arctic. The outer slopes of the mountains experience a significant softening effect of the climate of the surrounding plains, and the microclimate of the central part of the massif is much more severe. The mountains are covered with snow from October to June. The average annual temperature in the Khibiny is −2.5 °C. The average temperatures in January-February in the valley areas are –13 °C, in July—no higher than +13 °C. The air temperature drops with altitude decreasing by about 0.5 °C for every 100 m of altitude. The polar night lasts from 40 to 42 days. Cyclones and sudden changes in atmospheric pressure occur frequently. Hurricane winds can blow at speeds up to 50 m/s in the open spaces of the peaks. Summer is short, in the mountains it lasts 60–80 days. The period with an average daily temperature above 10 °C in the foothills lasts about 70 days. Summer also receives the maximum amount of precipitation. The polar day lasts 50 days. Precipitation ranges from 600–700 mm in the valleys to 1600 mm on the mountain plateaus of the Khibiny. Precipitation is distributed almost evenly during the year, a little more in summer, a little less in winter [15].

## 2.2. Geological Structure

The Khibiny alkaline massif is a large intrusive body of complex shape and composition. According to U-Pb, Rb-Sr, and Sm-Nd dating, the main types of rocks were formed from 380 to 360 million years ago (the Devonian period) [53]. A characteristic feature of the Khibiny massif is a ring (in plan) structure, which has a number of analogies among some other alkaline massifs [15,51,52]. The rock complexes making up the massif have the shape of arcs folded into each other and opening to the east due to the magma intrusion along the alternating ring and cone faults. The Khibiny alkaline massif consists of nephelinesyenite rocks, the rock-forming minerals of which are potassium–sodium feldspars,

aegirine $NaFe^{3+}(Si_2O_6)$ and nepheline $(Na,K)AlSiO_4$ [51]. As a result of the weathering rocks, nepheline is destroyed rather than feldspars, and alkali metal ions $Na^+$ and $K^+$ enter the lake in increased concentrations.

About 500 minerals have been found on the territory of the Khibiny massif, more than 100 of them have been discovered here, 110 are found nowhere else [52]. Many minerals have practical value. Apatite, nepheline, titanite, molybdenite and rinkite are or have been mined. Astrophyllite, aegirine, and eudialyte are used as ornamental stones. The world's largest deposits of apatite–nepheline ores are located on the territory of the Khibiny massif. The main minerals mined in the Khibiny are apatite, nepheline, sphene, aegirine, feldspar, titanomagnetite. Lovchorrite was previously mined. The massif contains the largest reserves of zirconium raw materials (zircon, eudialyte) and its accompanying hafnium (zircon), which are not currently mined. At the same time, a significant amount of this raw material is currently stored in the tailings of the apatite–nepheline factory.

According to the long-term plan, already in 1944, four mines were to operate in the "Apatite" Plant system: two apatite mines at Kukisvumchorr and Yuksporr, a sphene mine at Yuksporr, and a hibinite mine at Aikuaivenchorr. The ore they mined was to be processed by eight processing plants: two apatite, two nepheline, one sphene, one lovchorrite, one aegirine, and one rare earth. It was supposed to develop the production of phosphorus, to establish cement production [14]. However, the plan was not implemented due to the Second World War.

The following mines are currently operating: in the southeastern part of the Khibiny massif (catchment of the largest lake in the Murmansk region—Imandra)—Kirovsky (Kukisvumchorr and Yuksporr deposits) and Rasvumchorrsky (Apatite Circus and Rasvumchorr Plateau deposits) (Figure 1),in the southwestern part of the Khibiny massif (catchment of the deepest lake in the Murmansk region—Umbozero)—Vostochny (Koashva and Nyorkpakhk deposits) and the recently discovered Oleniy Ruchey (Koashva deposit) (outside of Figure 1). Mining is carried out both underground and open pit. The number of open pit mines is declining and soon the deposits will be developed only by the underground method.

In response to the plans of the North-Western Phosphorus Company, following the opening of the Oleniy Ruchey Mining and Processing Plant, to begin developing a new mine in the center of the Khibiny on the shores of Lake Goltsovoe (to date, forests have been cut down at the site of the future construction), a public environmental movement was formed advocating assigning the status of national park to the Khibiny and prohibiting their further development [54]. In 2018, the Khibiny alkaline massif was assigned the status of the Khibiny National Park (Decree of the Government of the Russian Federation of 8 February 2018 No. 130 "On the establishment of the Khibiny National Park". Date of access: 28 March 2018).

*2.3. Morphometric and Hydrological Characteristics of the Lake*

Bolshoy Vudjavr is a dam-type lake, the largest inland water body of the Khibiny (Figure 1). The lake is oval in shape and occupies a depression among the surrounding mountains in the northern taiga zone [15]. The soil cover is sparse, and mountain–tundra vegetation prevails in the catchment area of the lake. Mountains are absent only in the southern part of the lake. Here, the depression is closed by a ridge of sediments of the terminal moraine, cut through by a narrow valley flowing in the southwest of the lake by the Belaya River, which inflows into Lake Imandra. The lake is fed by surface and underground runoff. The absolute elevation of the lake is 312 m, the area of the water surface is 3.9 $km^2$, and the volume of water is 0.0579 $km^3$. The amplitude of fluctuations in the water level in the lake does not exceed 1.1 m (from 312.2 to 311.3 m above sea level). The maximum level is observed in the flood, the minimum—at the end of winter. River Juksporrjok (discharge 0.24–12 $m^3/s$) flows into Lake Bolshoy Vudjavr from the northeast, and River Vudjavrjok flows from the northwest (discharge 0.05–8 $m^3/s$). In the distribution of depths, there is a depression with a depth of 37.6 m in the northeastern part of the lake. The total annual amount of atmospheric precipitation in the catchment area is in the range

of 570–1060 mm, 820 mm on average. The temperature regime of Lake Bolshoy Vudjavr is determined by an ice-free period lasting 5–6 months. Studying the thermal regime of the lake showed that in summer there is a direct temperature stratification (from June to October), and in the winter—the reverse. Full spring and autumn homothermy usually sets in June and October, respectively. In winter, the temperature in the surface water layer is 0.4 °C, and is at a depth of 34 m –2 °C. In summer, the temperature of the surface layer on warm days can reach 15 °C.

*2.4. History of Development and Anthropogenic Activity*

The industrial development of the Khibiny began with the construction of the Murmansk railway during the First World War, which subsequently enabled exploring the geology and developing the mineral resources of the region. The geological surveys under the guidance of Academician A.E. Fersman in the early 1920s resulted in the discovery of the world's largest apatite–nepheline deposits with huge reserves of phosphorus raw materials in the Khibiny mountain range [51]. In 1929, the Production Trust "Apatite" was organized to develop the mineral resources of the Khibiny. Since 1931, processing apatite ore began at the processing plant (ANOF-I) on the shore of Lake Bolshoy Vudjavr; since 1934 the second stage was launched at this enterprise. From that time on, sewage began to flow into Lake Bolshoy Vudjavr and further along River Bolshaya Belaya without treatment. Municipal sewage from City of Kirovsk was also discharged without treatment. The apatite industry steadily increased its capacity and by 1938 produced more than 1 million tons of apatite concentrate [14]. The waters of the Juksporrjok River and further Lake Bolshoy Vudjavr were enriched with nitrogen and phosphorus compounds and silicic acid [55]. There is a sedimentation basin for wastewater from the Kirovsky and Rasvumchorrsky mines in the northeastern part of the lake, which comes with the flows of Rivers Juksporrjok and Saamskaya, fenced off from the main lake water area by a dam (Figure 1). Population growth and inefficient treatment facilities have led to the pollution of the lake by municipal wastewater from City of Kirovsk, as well as the Polar Alpine Botanical Garden. The ichthyofauna of Lake Bolshoy Vudjavr, numbering 5 species of fish, had practically disappeared by 1931 due to the pollution of the Bolshaya Belaya River, along which fish ascended to spawn from Lake Imandra. At present, the lake is inhabited by Arctic char and nine-spined stickleback [23], as well as European smelt, which has been actively migrating from Lake Imandra in recent years [24].

The chemical composition of the lake sediments depends on the natural (geochemical, hydrological, climatic, hydrobiological) features of the watershed and the lake itself, as well as direct and aerotechnogenic pollution of mining and metallurgical enterprises ("Apatite" Joint-Stock Company and the Kola Mining and Metallurgical Company (KMMC), a subsidiary of "Norilsk Nickel" Joint-Stock Company), as well as atmospheric transboundary transfers of pollutants. The influx of sewage and dust emissions from mines and ANOF-I of "Apatite" JSC, containing mainly nepheline and apatite $Ca_5(PO_4)_3(F,OH,Cl)$, enrich the water and sediments of the lake with Na, K, Ca, P, Al, Sr, F, and other elements that are part of apatite–nepheline ore and products of their processing. The Khibiny deposits are represented mainly by fluorapatite, but there are also other varieties of it. Apatite also contains impurities of Mn, Fe, Th, rare earth elements (REE), calcium carbonate $CaCO_3$ (carbonateapatite), and other impurities [51].

Atmospheric emissions from metallurgical plants can be transported over a distance of several tens or even hundreds of kilometers [56,57]. The Severonickel Copper andNickel Plant (Monchegorsk site of KMMC), located 45 km northwest of Lake Bolshoy Vudjavr, specializes in the production of electrolytic Ni and Cu, cobalt concentrate, sulfuric acid, and other products. Minerals such as pentlandite $(Fe,Ni)_9S_8$, chalcopyrite $CuFeS_2$, cobaltite $(Co,Ni)AsS$, and nickeline $NiAs$ are involved in the process of enrichment and metallurgical processing [58]), so all the above mentioned elements are present in high concentrations.

*2.5. Research Methods, Sampling and Analysis of Sediments*

2.5.1. Sediment Sampling

Sediment cores of 23 and 27 cm thickness were sampled on April 26, 2018 in the central part of Lake Bolshoy Vudjavr at a depth of 37.4 m (Figure 1). Sampling was carried out using the open gravity corer Limnos from the ice cover, allowing to extract stratified sediment core samples up to 60 cm long and separate these into 1 cm or thicker layers. After sampling, undisturbed sediment cores were transported to the laboratory, where they were divided into layers of 1 cm to determine the total content of elements and into layers of 5 cm (0–5, 5–10, 10–15 and 18–23 cm) to analyze the element fractions.Then, the samples were placed in a refrigerator, in which they were stored until dry at a temperature of about 4 °C according to the methodological guidelines [59–63]. For further study, the sediment samples were dried to an air-dry condition at room temperature and then to acompletely dry condition in an oven at a temperature of approximately 105–110 °C [64].

2.5.2. Granulometric Analysis of Sediments

The particle size distribution of sediments was determined using a multifunctional particle analyzer of the LS13 320 series (Beckman Coulter, Life Sciences, Indianapolis, IN, USA). This analyzer determines the particle size of sediments by laser diffractometry in water based on the physical principle of scattering of electromagnetic waves of different lengths. The technical features of the device allow the analysisof particles ranging in size from 0.04 μm to 2.0 mm in accordance with the ISO 13320-1 Standard.

2.5.3. Sediment Dating

When the collected sedimentsaccumulated in Lake Bolshoy Vudjavr, accordingly, the sedimentation rates were calculated from the change in the content of the $^{210}$Pb isotope with depth in the core (nonequilibrium $^{210}$Pb method) [65]. The model of constant initial concentration estimating only the average rate of sedimentation was used. The contents of $^{210}$Pb in individual layers of sedimentary core were determined from the alpha activity of its granddaughter isotope $^{210}$Po, which is in radioactive equilibrium with it and isolated from samples by the standard radiochemical method on nickel disks [66]. Sediment samples were dried at room temperature to a constant weight, followed by grinding in an agate mortar to a powdery state. The sample was placed in a solution of 15 mLof 7 M $HNO_3$ and kept for a day, and then was evaporated to dryness and treated with 15 mL of concentrated HF to completely dissolve the silicate fraction of the precipitate. The procedure was repeated thrice, each time evaporating the solution to dryness. At the next stage, the resulting residue was converted into the nitric acid form and evaporated again, after which the sample was dissolved in 100 mLof 0.1 M HCl with the addition of 2 mL of concentrated $H_2O_2$ with gentle heating until oxygen evolution ceased. By adding 100 mg of ascorbic acid, trivalent forms of iron were reduced and $H_2O_2$ residues were decomposed. Further, a pre-prepared nickel disk was placed into the resulting solution, fixed in a special fluoroplastic cassette, which ensures the release of $^{210}$Po on one side of the disk, and kept under stirring with a magnetic stirrer and 65 °C for 3 h. Further, the solution was poured off, and the removed disk was washed with distilled water and dried at 100 °C. The measuring targets prepared in this way were placed in radiometric equipment (an ORTEC Alpha-Duo spectrometer with an ion-implanted silicon detector) to obtain alpha spectra, which were used to measure the $^{210}$Po activities in each sample, and from them, the $^{210}$Pb concentrations were determined.

2.5.4. Determination of Loss on Ignition

Loss on ignition (LOI), which characterizes the content of organic matter in sediments (organic compounds are burnt in the temperature range from 100 to 500 °C), was determined by the gravimetric method after igniting an absolutely dry sample of each layer at a temperature of 550 °C in a muffle furnace to constant mass [60].

### 2.5.5. Chemical Analysis of Elements

The content of the main rock-forming elements (oxides of Al, Ca, Fe, K, Mg, Mn, Na, P, Ti, Si) in lake sediments was determined using an ARL ADVANT'X X-ray fluorescence spectrometer (Thermo Fisher Scientific). Before X-ray fluorescence analysis, the sediment samples were dried in an oven at a temperature of 105 °C to a constant weight.

To estimate the total concentrations of trace elements in lake sediments, open acid decomposition of sediment samples was carried out using HF, $HNO_3$, and HCl. Analytical weights 0.1 g of samples were used for the analysis. As a blank sample, a mixture of decomposition acids was used that underwent the same sample preparation procedure as sediment samples and a standard (control) sample—the chemical composition of the bottom silt of Lake Baikal BIL-1—GSO 7126-94. The concentrations of trace elements in sediment samples were determined by mass spectrometry on an XSeries-2 ICP-MS instrument. This technique is described in detail in a previously published article [63].

All laboratory work was based in the analytical center of the Institute of Geology of the Karelian Scientific Center of the Russian Academy of Sciences (Petrozavodsk). Programs Statistica (version 10), Microsoft Office Excel 2016, ArcGIS for Desktop 10.4.1 were used for statistical data processing and illustrations.

### 2.5.6. Determining the Chemical Fractions of Elements

To determine the fractions of various elements, a number of sequential extraction methods for soils have been adapted for sediments [67–70]. The schemes differ in the number of isolated fractions, as well as in the reagents used. The method for sequential extraction of element forms was used in this work, including the identification of the following:

- Water-soluble fractions (reagent $H_2O$).
- Available (exchangeable) fractions (reagent $NH_4CH_3COO$, pH 4.8). The elements can desorb in case of changing of pH or ionic composition of water.
- Fractions bound to Fe and Mn hydroxides (reagents 0.04 M $NH_2OH$+HCl in 25% $CH_3COOH$). Iron and manganese oxides exist in sediments as nodules, concretions, cement between particles. They are excellent scavengers for metals. The fraction is thermodynamically unstable under anoxic conditions (low Eh).
- Fractions bound to organic matter (reagents 0.02 M $HNO_3$ + 30% $H_2O_2$ and 3.2 M $NH_4CH_4COO$ in 20% $HNO_3$). Trace metals may be associated to various forms of organic matter. The complexation ability of humic substances in natural ecosystems is studied previously [71]. Organic matter can be degraded under oxidizing conditions leading to a release of trace metals in water.
- Acid-soluble (residual) fractions (reagent $HNO_3$); the most stable (mineral) fraction. The metals are associated with residual compound and not available to living organism. Dissolution of the fraction is possible only under a very strong anthropogenic impact on the reservoir.
- Mineral (silicate) fractions obtained by deducting the combined concentration of all the above fractions from the total concentrations. The metals bounded with the crystal structure of silicate minerals are not expected to be released in water over a reasonable time span under the conditions normally encountered in nature.

We decomposed the sediment samples to determine the total concentrations of heavy metals in the sediment in the core samples, and subdivided them into 5 cm layers by an acid breakdown using HF, $HNO_3$, and HCl in an open system.

### 2.5.7. Intensity of Lake Pollution

L. Håkanson's method [64] was chosen to assess the contamination of the sediments of Lake Bolshoy Vudjavr. To identify the intensity of contamination by substances, the values of the contamination factor ($CF^i$) were determined:

$$CF^i = C_m^i / C_n^i, \tag{1}$$

where $C_m^i$ is the content of the element in the layer m (cm) of sediments, $C_n^i$ is the pre-industrial value for this element, determined as the background value for the sediments of Lake Bolshoy Vudjavr in its deepest part (24–27 cm), deposited before the start of the work of the "Apatite" Plant, i.e., until 1929. This approach adhered to the following classification: $CF^i < 1$—low contamination factor (indicating low contamination of sediments by the studied element); $1 \leq CF^i < 3$—moderate contamination factor; $3 \leq CF^i < 6$—significant contamination factor; $CF^i \geq 6$—high contamination factor.

The ecological interpretation of the geochemical data was based on calculating the integrated Pollution Load Index (PLI) [72], which takes into account the content of all the studied elements in the lake sediment core:

$$PLI = (CF_1 * CF_1 * CF_n)^{1/n}, \qquad (2)$$

where $CF_n$ is the contamination factor (the ratio of element concentration to the background value of this element). Depending on the value of the PLI, the pollution level can be assessed as low (PLI $\leq$ 1), moderate (1 < PLI < 2), high (2 < PLI < 3), and extremely high (3 < PLI).

## 3. Results and Discussion

### 3.1. Granulometric Composition of Sediments

Analysis of the granulometric composition of the sediments of Lake Bolshoy Vudjavr revealed that fractions smaller than 0.05 mm (88.4% of the total mass of the sample) prevail in the sediments and fractions in the range from 0.05 to 2 mm account for 11.6% of the total mass of the sample. Thus, the studied sediments can be attributed to the silty type of lacustrine sediments [73]. The particle size distribution was previously assessed by the pipette method based on the difference in the settling rates of particles of different sizes in water, the prevalence of smaller size fractions (clayey less than 0.005 mm) in the sediments of Lake Bolshoy Vudjavr was revealed and the deposits were classified as clayey [17]. This is probably due to different methods for analyzing the particle size distribution of sediments in these studies.

### 3.2. Dating of Sediments

As mentioned above, the distribution of the $^{210}$Pb isotope along the profile of the sampled sediment core was used to date the sediments of Lake Bolshoy Vudjavr. The decrease in the $^{210}$Pb content as a whole has a consistent character, reaching a basic value at a depth of about 22 cm, which almost does not change further with depth (Figure 2). This allows concluding that the excess of $^{210}$Pb ($^{210}$Pb$_{ex.}$ in Figure 2) in the upper seven studied layers of the lacustrine sediment core is not supported by radioactive equilibrium, and the concentrations determined for the three lower layers are considered to be maintained by the equilibrium in the $^{238}$U series. Therefore, their weighted average value (39.11 $\pm$ 0.59 Bq/kg) can be applied to calculate the concentration of $^{210}$Pb$_{ex.}$ in the overlying layers. The data on the content of the $^{210}$Pb$_{ex.}$ isotope in the layers from 0 to 20 cm are well approximated by the exponent (r = $-$0.852), which is obviously due to the decrease in its concentration according to the law of radioactive decay with a half-life of $T_{1/2}$ = 22.2 years at a relatively constant sedimentation rate, which means that it allows applying the calculation model of constant initial concentration [65]. Calculations using this model showed that the sediment layer within the horizon of 2–20 cm accumulated over 75.23 $\pm$ 2.11 years, and the average sedimentation rate over the calculated period of time can be estimated as 2.26 $\pm$ 0.06 mm/yr. Notably, the obtained sedimentation rate is close to the sedimentation rates (2.5–3.0 mm/yr) in small lakes in the taiga, forest–tundra, and steppe landscapes of Siberia [74]. The similar sedimentation rate (1.5–3 mm/yr) was obtained in Lakes Imandra and Kuetsjärvi (northwestern Murmansk Region) affected by apatite–nepheline and copper–nickel production runoff [16,75]. At the same time, the calculated sedimentation rate in this work is higher than the average sedimentation rate (0.3-1.25 mm/yr) in the lakes of Northern Fennoscandia and Murmansk region, which are not directly polluted by industrial enterprises [26,62,76,77], and some lakes in the southern part of the Republic of Karelia [78].

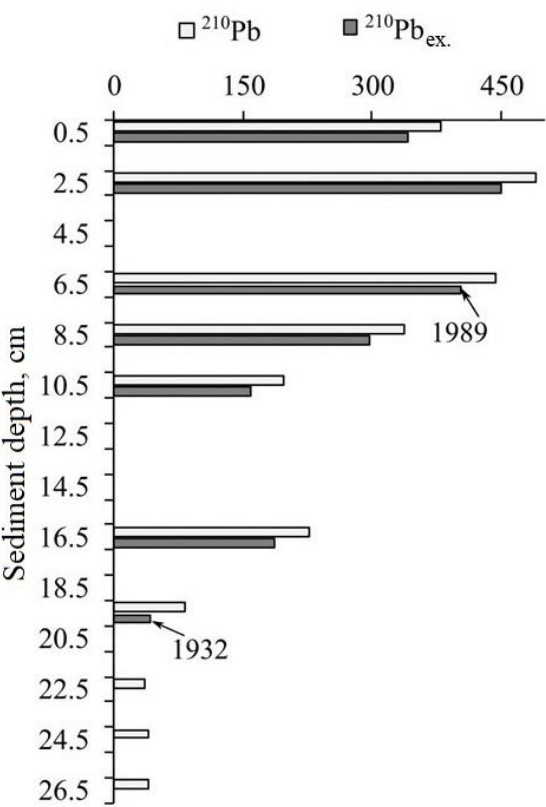

**Figure 2.** Contents of $^{210}$Pb and $^{210}$Pb$_{ex.}$ (Bq/kg) in separate layers of the sediment core from Lake Bolshoy Vudjavr with indication when the individual layers were formed, calculated from the average sedimentation rate.

### 3.3. The Main Rock-Forming Elements

The Khibiny massif is the world's largest intrusion of nepheline syenites, melteigite–urtites, and apatite–nepheline rocks with an area of more than 1300 km$^2$ [52]. The listed rocks, as well as the main part of other rocks that make up the Khibiny, are composed mainly of aluminosilicate minerals. Therefore, in the chemical composition of the studied sediments of Lake Bolshoy Vudjavr, oxides of Si (average content 44%), Al (16%), Ca (5.9%), and Fe (5.3%) prevail among the main components (Figure 3). In addition, according to the LOI, it can be concluded that the content of total organic matter in the sediments is 17.4%. According to the classification given in the work of D.A. Subetto [73], these deposits should be referred to siliceous siltstone sapropels in terms of particle size distribution, content of silicon and organic material. The organic material content in the studied sediments can be considered rather low for the lakes of the taiga zone of Russia, which can be explained by the geographical location of the reservoir—inside the mountain range with sparse soil-vegetation cover and significant filtration capacity of the clastic material of the upper rock layers, where the main material coming from the watershed into the lakes is represented by a terrigenous component, as well as by the influx of a large amount of suspended mineral particles with apatite–nepheline production effluents from the mines and the processing plant. The organic matter content in the sediments of lakes located in the plain (boggy or forest) regions in the north of Russia most often varies from 30 to 80% [79]. A similar content of organic material was recorded in the sediments of lakes in the Murmansk region [80].

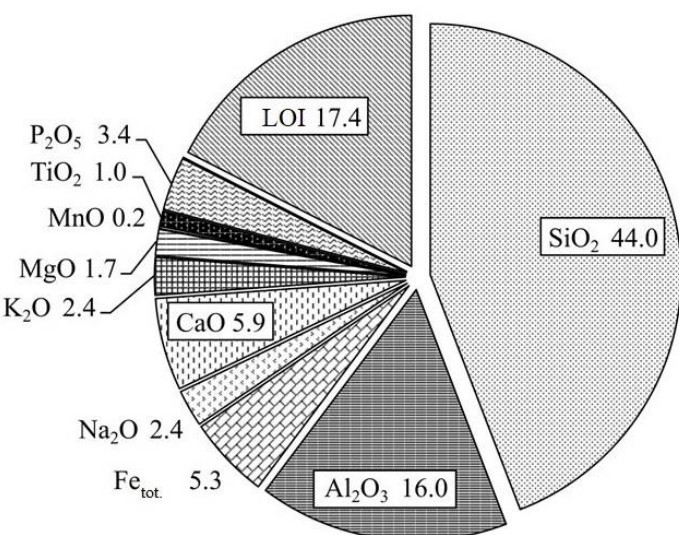

**Figure 3.** The average content (%) of the main elements and organic matter (LOI) in the modern sediments of Lake Bolshoy Vudjavr.

The macroelements composing apatite $Ca_5(PO_4)_3(F,Cl,OH)$ and nepheline $(Na,K)AlSiO_4$ are found in high concentrations in the surface sediment layers of Lake Bolshoy Vudjavr. On the one hand, the Si oxide content decreases along the core of the studied sediments of Lake Bolshoy Vudjavr from 51–52% in the lower layers to 38–39% towards the sediment surface (Figure 4, Table 1). On the other hand, there is a noticeable increase in the concentrations of P (by an order of magnitude) (Figure 5), Ca (by a factor of 3.5), K, Na, and Ti (the last three elements by more than a factor of 2) oxides, which, according to early works on this lake [18,19] is due to the extraction and processing of apatite–nepheline raw materials by the "Apatite" Plant since the 1930s. The organic material content (LOI) in the surface sediment layers decreases (Figure 4), which is associated with the suppression of biological production in the lake due to severe pollution and restructuring of hydrobionts, as well as the influx of a large amount of suspended mineral particles with wastewater from apatite–nepheline production [18,19].

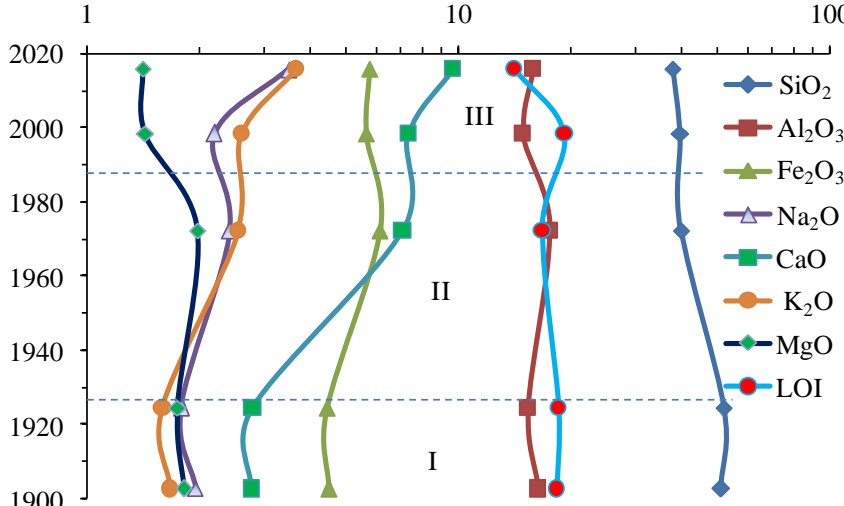

**Figure 4.** Vertical distribution of the content (%) of the main elements and organic matter (LOI) in the sediment core of Lake Bolshoy Vudjavr.

**Table 1.** The content of organic matter (LOI) and macroelements (in %) in apatite ore, tailings of apatite production, and surface and background layers of sediments in lakes Bolshoy Vudjavr and Imandra according to the results of X-ray fluorescence analysis.

| Elements | Apatite Ore[1] | Tailings[1] | Bolshoy Vudjavr | | Imandra[1] | |
|---|---|---|---|---|---|---|
| | | | 0–1 cm | 26–27 cm | 0–1 cm | 18–19 cm |
| LOI | – | – | 14.08 | 18.36 | 29.33 | 13.69 |
| $SiO_2$ | 26.6 | 39.8 | 37.68 | 51.09 | 69.43 | 75.27 |
| $Al_2O_3$ | 12.1 | 17.7 | 15.87 | 16.35 | 9.51 | 9.98 |
| $Fe_2O_3$ | 6.49 | 8.14 | 5.74 | 4.45 | 6.41 | 3.99 |
| $Na_2O$ | 6.23 | 9.74 | 3.48 | 1.95 | 0.95 | 1.77 |
| CaO | 24 | 9.72 | 9.68 | 2.77 | 2.98 | 2.31 |
| $K_2O$ | 2.83 | 4.28 | 3.65 | 1.67 | 1.01 | 1.07 |
| MgO | 1.27 | 1.52 | 1.41 | 1.82 | 0.80 | 1.72 |
| MnO | 0.188 | 0.247 | 0.19 | 0.12 | 0.92 | 0.14 |
| $TiO_2$ | 3.01 | 4.07 | 1.33 | 0.62 | 0.46 | 0.41 |

Notes: [1] [81,82].

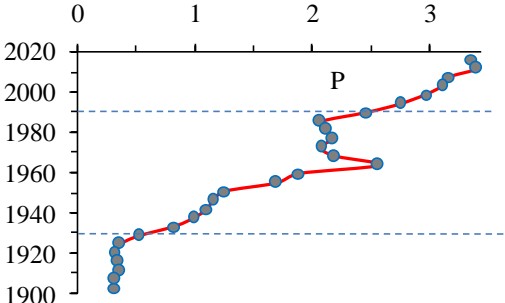

**Figure 5.** Vertical distribution of phosphorus content (%) in the sediments of Lake Bolshoy Vudjavr.

A significant increase in P concentrations in the studied sediment core begins in the 20–21 cm layer, which corresponds to the beginning of the 1930s, and steadily increases from 0.3 to 3.4% in the upper layers of the lacustrine sediments (Figure 5). The noticeable increase in the P content can be traced in the upper part of the core after the collapse of the Soviet Union and the onset of the economic crisis after the 1990s, although the production at the "Apatite" Plant decreased by more than three times. Perhaps this is due to the decrease in the enterprise's expenses for environmental protection during this period and the decrease in the efficiency of sedimentation of suspended matters in a fenced dam on the lake.

The content of the main rock-forming elements in the sediments of Lake Bolshoy Vudjavr exceeds their concentrations in the surface and background layers of the sediments of Lake Imandra (with the exception of Si, Fe, and Mn), which is associated with the influence of the runoff of the "Apatite" Plant (in the surface sediment layers) and the geochemical features of the territory catchment area (background sediment layers) in Lake Bolshoy Vudjavr (Table 1). Furthermore, the content of many main elements (Si, Al, Ca, K, Mg) in the surface part of the sediments is comparable to their content in the tailings of apatite–nepheline ore processing, and the P content is even higher than in the tailings (Table 1).

Based on studying the content of the main rock-forming elements, the thickness of lake sediments was divided into threelayers corresponding to the stages of formation of the chemical composition of sediments—the pre-industrial period (Figure 4,layer I), after the start of development of apatite–nepheline deposits and a constant increase in production

(layer II), the current state of the lake (layer III). The vertical distribution of elements in the sediments of the lake will be further described according to these stages of development of the enterprise and sedimentation.

### 3.4. Trace Alkali and Alkaline Earth Metals

Trace alkali and alkaline earth metals were called metals, which, unlike the main rock-forming metals (Na, K, Mg, Ca), are found in the sediments of Lake Bolshoy Vudjavr in much smaller quantities, but in increased contents relative to other lakes in Murmansk region, both in surface and in the background layers (Table 2). The concentrations of trace alkali and alkaline earth metals in the sediment core of Lake Bolshoy Vudjavr have sharply increased since the extraction and processing of apatite–nepheline raw materials started in 1930s (Figure 6). The content of trace alkali and alkaline earth metals also increased after the collapse of the Soviet Union and the economic crisis of 1990, which is associated with an increase in production in the 2000s. The content of all alkali and alkaline earth metals ($r = 0.87$–$0.97$) was closely correlated, exception for Li, whose distribution differed from other alkali and alkaline earth metals. Nevertheless, even this lightest metal sharply increased its content in the early 1930s, when the development of apatite–nepheline ores started (Figure 6). Lithium isomorphically replaces potassium in widespread rock-forming minerals, and it is also found in some minerals of the Khibiny massif, for example, manganneptunite $KNa_2LiMn^{2+}_2Ti^{4+}_2[Si_8O_{24}]$. The largest increase in the content in the surface layers relative to the background values was noted for Sr (by a factor of 8), while the concentrations of other trace alkali and alkaline earth metals increased not so significantly (from 1.5 to 2.7) (Figure 6, Table 2).

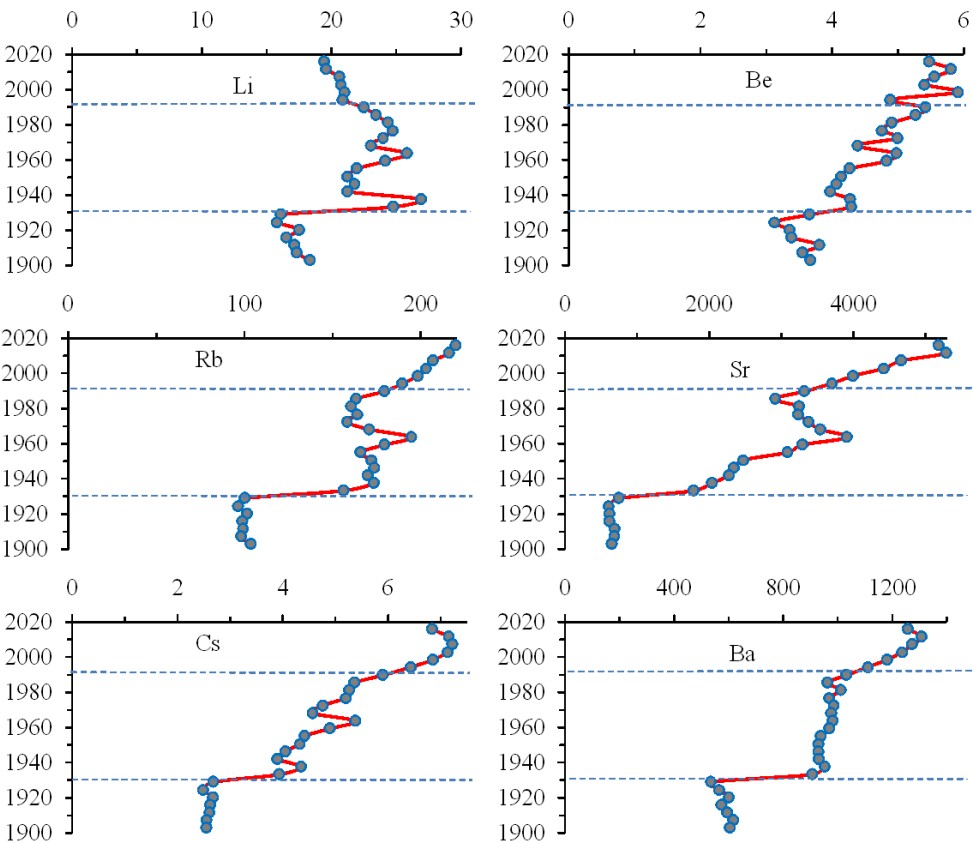

**Figure 6.** Vertical distribution of the content (µg/g) of alkali and alkaline earth metals in the sediments of Lake Bolshoy Vudjavr.

**Table 2.** The content of trace elements (in µg/g) in apatite ore, tailings of apatite production, and surface and background layers of sediments of lakes Bolshoy Vudjavr, Imandra, Rybachy, and Severnoye according to the results of mass spectrometry with inductively coupled plasma (ICP-MS).

| Element | Apatite Ore [1] | Tailings [1] | Bolshoy Vudjavr | | Imandra [1] | | Rybachy [2] | | Severnoye [3] | |
|---|---|---|---|---|---|---|---|---|---|---|
| | | | 0–1 cm | 26–27 cm | 0–1 cm | 18–19 cm | 0–1 cm | 26–27 cm | 0–1 cm | 37–39 cm |
| Li | 4.41 | 6.45 | 19.46 | 18.34 | 15.1 | 13.4 | 13.2 | 8.70 | 12.63 | 5.07 |
| Be | 3.21 | 6.39 | 5.48 | 3.67 | 1.84 | 0.75 | 1.30 | 1.49 | 0.64 | 0.49 |
| P | 73759 | 16093 | 33590 | 3213 | – | – | – | – | 3488 | 1128 |
| Sc | 2.52 | 0.76 | 9.15 | 17.05 | 2.62 | 7.96 | 4.91 | 5.22 | 12.18 | 9.76 |
| Ti | 719 | 1011 | 9109 | 3932 | – | – | 1733 | 1362 | 1257 | 479 |
| V | 261 | 303 | 137 | 119 | 94.5 | 109 | 63.1 | 53.5 | 528 | 40 |
| Cr | 32.8 | 33.7 | 47.31 | 87.79 | 94.9 | 112 | 61.5 | 51.1 | 58.49 | 34.77 |
| Mn | 234 | 400 | 1624 | 964 | – | – | 1932 | 203 | 174 | 114 |
| Co | 19.6 | 17.6 | 19.15 | 15.43 | 29.6 | 16.3 | 40.0 | 3.98 | 20.08 | 13.85 |
| Ni | 13.4 | 9.38 | 32 | 38 | 1751 | 59 | 127 | 24 | 171 | 27 |
| Cu | 79.1 | 101 | 225 | 55 | 429 | 96 | 370 | 38 | 111 | 25 |
| Zn | 47 | 71 | 245 | 126 | 132 | 78 | 148 | 56 | 416 | 64 |
| Ga | 26.4 | 39.9 | 27.57 | 64.44 | 27.7 | 30.8 | 7.52 | 5.90 | – | – |
| As | 2.80 | 0.86 | 3.78 | 6.11 | 19.1 | 9.1 | 6.28 | 1.89 | – | – |
| Rb | 68 | 90 | 220 | 104 | 49.8 | 37.4 | 32.4 | 18.7 | 19.28 | 3.83 |
| Sr | 13900 | 3710 | 5186 | 652 | 941 | 187 | 153 | 169 | 157 | 53 |
| Y | 161 | 72.9 | 104 | 30 | 40 | 21 | 17 | 36 | 10 | 11 |
| Zr | 317 | 584 | 606 | 645 | 163 | 123 | 53 | 45 | 26 | 11 |
| Nb | 168 | 288 | 191 | 108 | 36 | 8.5 | 5.4 | 3.7 | 3.0 | 1.2 |
| Mo | 6 | 6.53 | 8 | 10 | 24 | 11.4 | 6.1 | 11.4 | 8.3 | 23.5 |
| Cd | 0.36 | 0.081 | 0.71 | 0.57 | 1.29 | 0.34 | 0.50 | 0.45 | 1.18 | 0.30 |
| Sn | <24 | <24 | 3.90 | 2.17 | 16.30 | 13.50 | 32.1 | 0.61 | 7.11 | 0.80 |
| Sb | 85 | 70 | 4.82 | 0.29 | – | – | 0.50 | 0.138 | 2.68 | 0.09 |
| Cs | – | – | 6.84 | 2.55 | – | – | 1.98 | 0.91 | 0.83 | 0.25 |
| Ba | 603 | 1100 | 1257 | 606 | 685 | 486 | 198 | 169 | 258 | 78 |
| La | 1285 | 280 | 535 | 89 | 136 | 57 | 29 | 70 | 26 | 20 |
| Ce | 2130 | 460 | 779 | 153 | 236 | 81 | 50 | 98 | 39 | 27 |
| Pr | 190 | 44 | 79.2 | 15.4 | 24.8 | 13.6 | 6.6 | 15.6 | 5.1 | 4.9 |
| Nd | 640 | 154 | 283 | 54 | 95 | 53 | 25 | 62 | 19 | 19 |
| Sm | 81.3 | 21.6 | 43.6 | 9.0 | 13.6 | 7.7 | 5.17 | 11.63 | 3.23 | 3.31 |
| Eu | 24.6 | 7.1 | 12.21 | 2.38 | 3.91 | 1.83 | 1.08 | 2.41 | 0.75 | 0.71 |
| Gd | 98.1 | 26.8 | 34.8 | 7.0 | 14.10 | 7.25 | 3.84 | 8.92 | 2.73 | 2.78 |
| Tb | 8.16 | 2.51 | 4.61 | 1.07 | 1.46 | 0.86 | 0.56 | 1.31 | 0.36 | 0.38 |
| Dy | 33.1 | 11.4 | 22.17 | 5.69 | 8.68 | 4.11 | 2.98 | 6.78 | 1.74 | 1.92 |
| Ho | 5.33 | 1.96 | 3.70 | 0.99 | 1.43 | 0.69 | 0.53 | 1.11 | 0.31 | 0.36 |
| Er | 11.40 | 4.64 | 9.09 | 2.96 | 3.45 | 1.96 | 1.73 | 3.36 | 0.87 | 1.04 |
| Tm | 1.22 | 0.545 | 1.06 | 0.39 | 0.32 | 0.30 | 0.20 | 0.38 | 0.12 | 0.13 |
| Yb | 6.04 | 2.79 | 5.55 | 2.44 | 2.57 | 1.76 | 1.34 | 2.62 | 0.69 | 0.88 |
| Lu | 0.61 | 0.309 | 0.61 | 0.30 | 0.27 | 0.24 | 0.20 | 0.35 | 0.10 | 0.12 |

**Table 2.** *Cont.*

| Element | Apatite Ore [1] | Tailings[1] | Bolshoy Vudjavr | | Imandra[1] | | Rybachy[2] | | Severnoye[3] | |
|---|---|---|---|---|---|---|---|---|---|---|
| | | | 0–1 cm | 26–27 cm | 0–1 cm | 18–19 cm | 0–1 cm | 26–27 cm | 0–1 cm | 37–39 cm |
| Hf | 0.73 | 0.328 | 9.35 | 11.08 | 4.36 | 4.10 | 1.69 | 1.34 | 0.59 | 0.31 |
| Ta | 0.61 | 0.173 | 7.34 | 3.82 | 2.12 | 0.64 | 0.19 | 0.09 | 0.23 | 0.06 |
| W | <24 | 34.5 | 4.00 | 2.82 | 38.10 | 31.4 | 0.78 | 0.33 | 8.70 | 1.00 |
| Hg | 0.051 | 0.024 | 0.308 | 0.050 | 0.282 | 0.034 | – | – | – | – |
| Tl | – | – | 0.12 | 0.19 | – | – | 0.14 | 0.09 | 0.49 | 0.29 |
| Pb | 2.66 | 2.99 | 24.6 | 9.9 | 28.0 | 4.5 | 82.0 | 10.7 | 44.0 | 6.3 |
| Bi | – | – | 0.35 | 0.10 | – | – | 0.98 | 0.058 | 0.24 | 0.07 |
| Th | 13.7 | 14.9 | 23.8 | 11.3 | 8.0 | 7.9 | 4.7 | 6.4 | 2.3 | 1.2 |
| U | 2.76 | 3.25 | 12.8 | 9.4 | 8.0 | 15.1 | 44 | 190 | 77 | 204 |

Notes: [1] [81,82], [2] [27], [3] [83].

Feldspars (rock-forming minerals of the Khibiny rocks) in addition to potassium, sodium, and calcium also contain other alkali and alkaline earth metals. Feldspars are used as a raw material to extract rubidium. Potassium-barium feldspars (hyalophanes) are known. The rocks of apatite–nepheline deposits contain a large number of minerals made up of trace alkali and alkaline earth metals, for example, varieties of the mineral pyrochlore $(Na,Ca)_2Nb_2O_6(OH,F)$ widespread in the Khibiny—bariopyrochlore $(Ba,Sr)_2(Nb, Ti)_2(O,OH)_7$, strontium pyrochlore $Sr_2Nb_2O_7$, as well as gadolinite $Ce_2Fe^{2+}Be_2[SiO_4]_2O_2$, belovite $Sr_3Na(Ce,La)[PO_4]_3F$, barite $Ba[SO_4]$, ankylite $Sr(Ce,La)[CO_3]_2(OH)\cdot H_2O$, strontianite $Sr[CO_3]$, burbankite $(Na,Ca)_3(Sr,Ba,Ca,REE)_3[CO_3]_5$, and many others [52].

The main ore mineral fluorapatite contains a large amount of strontium (SrO up to 10%) and other alkali and alkaline earth metals. There is even a variety of fluorapatite, strontioapatite $Sr_3Ca_2[PO_4]_3F$, in which the content of strontium oxide is about 50% [52]. The correlation coefficients of trace alkali and alkaline earth metals (except for Li) with phosphorus, the main element of the mineral apatite, are very high (r = 0.91–0.98, except for Li). Therefore, the main source of these metals in the sediments of Lake Bolshoy Vudjavr is probably mining and processing effluents of apatite raw material, although numerous trace alkali and alkaline earth metal minerals recovered from the mines during ore extraction also supply a certain amount of these trace metals to the sediments.

*3.5. Rare Earth Elements (REE)*

REE are a group of 17 elements including scandium, yttrium, and the lanthanides (from lanthanum to lutetium). The anthropogenic factors (industrial emissions and effluents, as well as dust from construction, deterioration of residential and road structures, weathering of soils, and Quaternary sediments in urban areas) play an important role in the accumulation of REE in the lake sediments in industrial areas and urban water bodies [83]. Environmental pollution with REE is associated with the activities of mining companies, industrial enterprises, as well as with the combustion of coal containing impurities of various metals, including REE, at energy enterprises [84].

An increase in the concentrations of all REE from two to six times is recorded in the upper sediment layers of Lake Bolshoy Vudjavr (the largest increase was noted for light REE, for example, for La—6.0), which results from the extraction and processing of apatite–nepheline ores (Figure 7). A similar distribution of REE was recorded in the sediments of Lake Imandra in the zone of wastewater inflow of the "Apatite" Plant [49,81,82], but sediment layer with elevated REE contents in Lake Imandra is two times smaller (10 cm), which indicates a lower sedimentation rate in it. There is an increase in the content of REE in the mid-1960s, and then a decrease in the concentration of REE, which is probably associated with the commissioning of the ANOF-I tailing dump and a decrease in the anthropogenic load on the lake. A large amount of REE is contained in fluorapatite ($Y_2O_3$ up to 3.3%, $La_2O_3$ up to 1.7%, $Ce_2O_3$ up to 4.1%, $Nd_2O_3$ up to 1.4%, etc.) [52]. There are also numerous

minerals containing REE [52], for example, lovchorrite $(Na(Ca,Na)_2(Ca,Ce)_4TiO_2F_2(Si_2O_7)_2$ (REE oxides $TR_2O_3$ up to 17%), in which calcium can be replaced by Sr (up to 3.6%), Ce and U, loparite $(Ce,Na,Ca)_2(Ti,Nb)_2O_6)$ $(Ce_2O_3$ up to 20%, $La_2O_3$ up to 13%, $Nd_2O_3$ up to 3.7%), gadolinite $Ce_2Fe^{2+}Be_2[SiO_4]_2O_2$ $(Ce_2O_3$ up to 57%, $La_2O_3$ up to 18%, $Nd_2O_3$ up to 5%), nordite $Na_3Sr(Ce,La)Zn[Si_6O_{17}]$ $(Ce_2O_3$ up to 20%, $La_2O_3$ up to 10%), rinkite $Na_2Ca_4(Ce,Y)Ti[Si_2O_7]_2(F,OH,O)_4$ $(Ce_2O_3$ up to 11%, La2O3 up to 5%, $Nd_2O_3$ up to 4%), and cerite $(Ce,La,Ca)_9(Fe3+,Mg)[SiO_4]_6[SiO_3(OH)](OH)_3$ $(Ce_2O_3$ up to 30%, $La_2O_3$ up to 24%, $Nd_2O_3$ up to 5.6%).

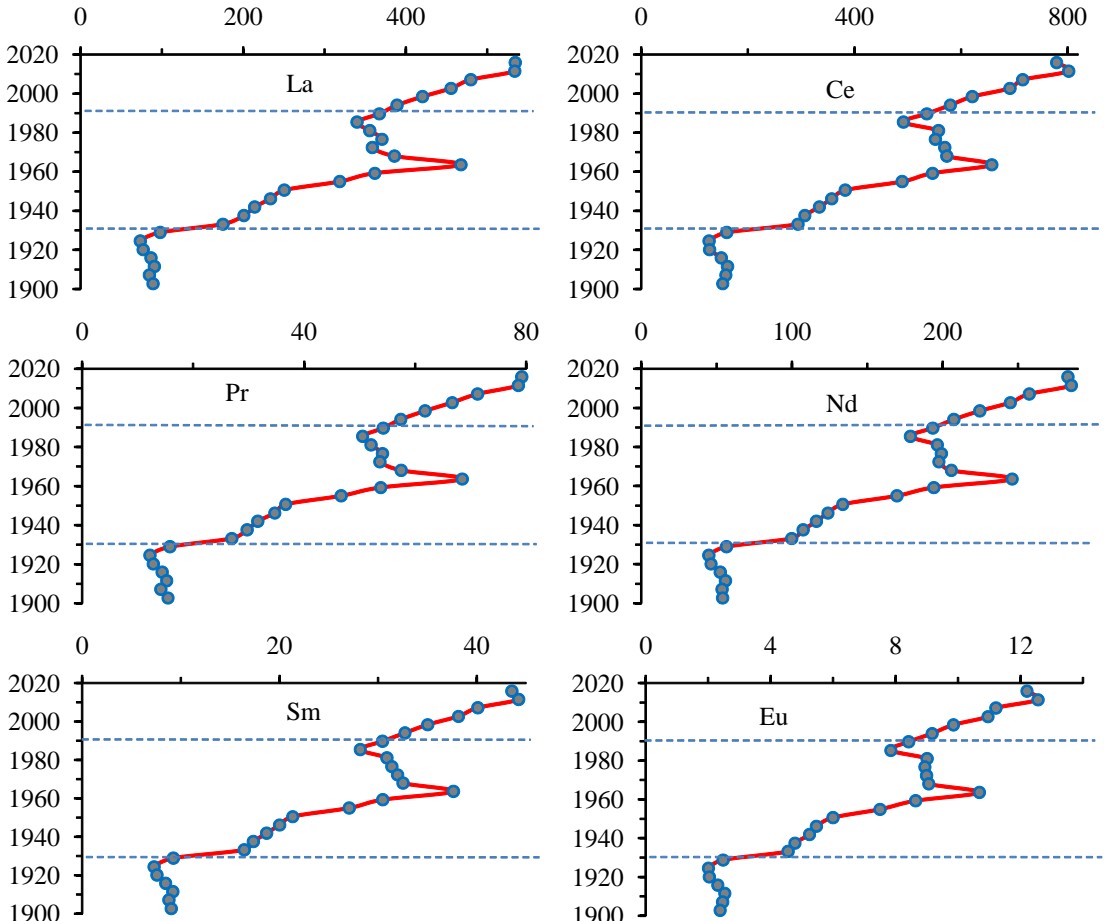

**Figure 7.** Vertical distribution of the content (μg/g) of light rare earth elements in the sediments of Lake Bolshoy Vudjavr.

The distribution profiles of all REE are very similar (except for Sc, the content of which decreases from background to surface layers), and the vertical distribution of the light REE content in the sediments of Lake Bolshoy Vudjavr are shown in Figure 7 as an example. A similar distribution of all REE (with the exception of Sc) is indicated by the correlation coefficient between REE, which in most cases, is equal to 1.00. Scandium with all REE shows a correlation coefficient ranging from –0.91 to –0.95.

The content of REE in the surface sediment layers of Lake Bolshoy Vudjavr is the highest of all known values for the lakes of Murmansk region (Table 2) and the Republic of Karelia [83], which indicates the inflow of effluents with a high content of REE from JSC "Apatite" into the lake. The background REE contents in the sediments of lakes Bolshoy Vudjavr and Rybachy are comparable (Table 2), which indicates the similarity of the natural geochemical setting of these two lakes in Murmansk region. The highest accumulation of REE took place in the sediments of lakes in the tundra zone, while the lowest accumulation of REE was noted in the lakes of the forest zone (taiga) [84]. Lake Bolshoy Vudjavr is

located in the mountain tundra zone, and Lake Rybachy is located in the flat tundra zone. In addition, REE-Th-U mineralization of pegmatoid granites within the Litsevsky uranium ore region, including the Rybachy and Sredny Peninsulas [85], is manifested in the elevated REE contents in the background sediments of Lake Rybachy.

*3.6. Trace Elements That Reduce the Content throughout the Sediment Core*

The Khibiny alkaline massif contains in its depths the largest reserves of zirconium raw materials (zircon $Zr[SiO_4]$, eudialyte $Na_{12}Ca_6Fe^{2+}{}_3Zr_3[Si_3O_9]_2[Si_9O_{24}(OH)_3]_2$, baddeleyite $ZrO_2$ [52]) and its accompanying hafnium which are currently not processed. In total, 3.7 thousand tons of eudialyte concentrate were planned to be produced at the "Apatite" Plant during the five-year plan 1933–1937 [14]. The increase in the content in the distribution profile of Zr and Hf in the sediments of Lake Bolshoy Vudjavr was recorded in the 1930–1940s (Figure 8), which indicates the consequences of the extraction of zirconium raw materials at that time. The content of Zr and Hf decreases to the sediment surface since the extraction of zirconium raw materials was terminated. The content of Zr and Hf correlates very closely with each other (r = 0.97).

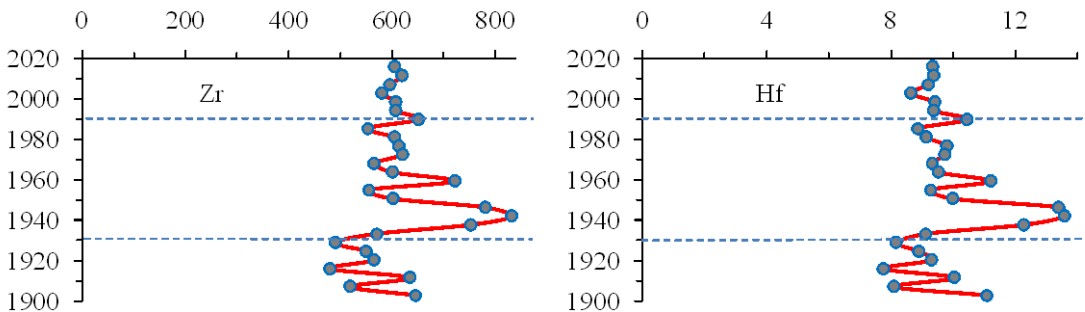

**Figure 8.** Vertical distribution of the content (µg/g) of Zr and Hf in the sediments of Lake Bolshoy Vudjavr.

The Mo minerals molybdenite $MoS_2$ molybdite $MoO_3$, wulfenite $PbMoO_4$, and fer-rimolybdite $Fe_2(MoO_4)_2 \cdot 8H_2O$ are fairly widespread in the Khibiny mountain range [52]. The conditions to find molybdenite were described in detail by A.N. Labuntsov, who discovered the Takhtarvumchorr molybdenite deposit in 1927 [80]. A decision was made in 1933 to build a molybdenum mine and a molybdenum processing plant [14]. Molybdenite was mined in the early 1930s at the Takhtarvumchorr mine, and then the mining stopped due to "insignificance of reserves and poverty of ore", as well as due to the discovery of more promising deposits of molybdenum ores in Kazakhstan and the Caucasus [86]. There is a significant increase in the content of Mo in the 1920s, which is most likely associated with the exploration and development of molybdenum ores from the Takhtarvumchorr deposit (Figure 9).

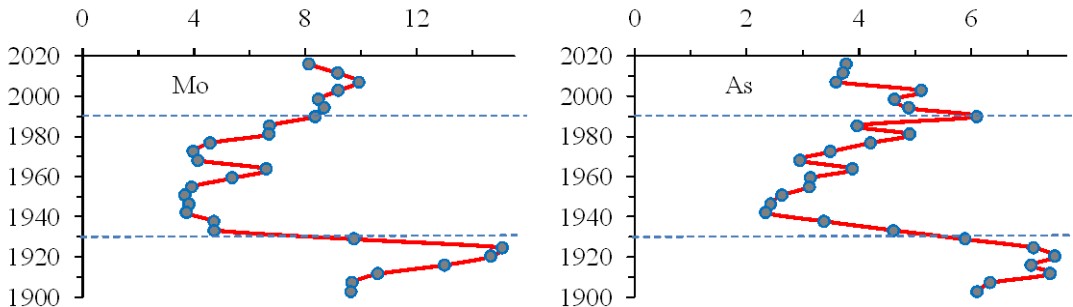

**Figure 9.** Vertical distribution of the content (µg/g) of Mo and As in the sediments of Lake Bolshoy Vudjavr.

The content of Mo closely correlates with that of As (r = 0.86), and their distributions in the sediments of Lake Bolshoy Vudjavr completely coincide (Figure 9). The rocks of the Khibiny alkaline massif contain a number of minerals containing arsenic—arsenopyrite FeAsS, cobaltite CoAsS, lellingite $(Fe,Co)As_2$, and safflorite $(Co,Fe)As_2$ [52]. Probably, these minerals are also common in the rocks of molybdenum ores of the Takhtarvumchorr deposit.

The concentrations of Cr, Ga, and Tl in the sediments of Lake Bolshoy Vudjavr significantly decreased after the development of apatite–nepheline deposits started, which is associated with their lower content in the ore and tailings (Table 2), and, accordingly, in the effluents of mines and the concentration plant.

*3.7. Trace Elements That Reduce the Content in the Upper Sediment Layers*

An increase in the concentrations of almost all the studied elements is observed in the vertical distribution of elements in the sediment core of Lake Bolshoy Vudjavr, starting from the layer of 20-21 cm, which indicates the period of the beginning of mining and processing of apatite raw materials (1930s). At the same time, a large group of elements stands out, the content of which has been decreasing in recent decades for various reasons (Figure 10).

The 10–15-fold excess of Sb in the uppermost layers of the studied sediments is especially distinguished, which makes this element one of the most potentially dangerous for the urban lake ecosystem. In general, this is a common phenomenon for urban lakes in Murmansk region and Karelia. Thus, Sb was noted as a priority pollutant for small lakes in the cities of Petrozavodsk and Murmansk [87,88], for example, a 30-fold increase in the content of Sb in the surface sediment layer compared to the background content (Table 2). Antimony is wellknown to accumulate in different environments as a result of the long-range transport of pollutants [89]; however, in the case of Lake Bolshoy Vudjavr, there may be a strong influence of a local coal-fired thermal power plant located 10 km from the reservoir. The enterprise began operation in the late 1950s and is still operating, which can be seen from the Sb distribution in the sediment core of Lake Bolshoy Vudjavr—an increase in the concentration of this metal is recorded just in the middle of the 20th century (Figure 10). As Sb is one of the most carboniferous metals [90], the coals used at the local thermal power plant are likely to have an increased content of this element.

Contamination of the studied sediments with V and Ni is also directly related to the heating supply of Kirovsk,until a boiler house began operating in the city in 2013, using masut as fuel. The content of V and Ni in the upper sediment layer reached the background values (Figure 10). Masut ash is known to be enriched in these elements, which enter the atmosphere during fuel combustion and then are washed into soil, water, and reservoir sediments [91,92]. Currently, the boiler house is not functioning, and the heating supply of Kirovsk is provided by the coal-fired thermal power plant in Apatity. An increase in the content of V and Ni was recorded in the sediments of Murmansk, which is associated with the influence of masut boilers and thermal power plants of the city, which began to operate on this type of fuel in the 1960s (coal was previously used) [93]. A distribution similar to these metals in the sediments of Lake Bolshoy Vudjavr was noted for W (Figure 10), which closely correlates with V and Ni (r = 0.92). Abnormally high contents of W and Sb were found in the coal samples obtained from the deposits of the Northern Urals [94].

The content of U in the sediments of Lake Bolshoy Vudjavr sharply increased in the 1970s and 1980s, possibly due to the peaceful nuclear explosions of Dnepr-1 (4 September 1972) and Dnepr-2 (27 August 1984) at the Kuelporsky mine to increase the fragmentation of the ore body and reduce the cost of extracting the ore. It is quite probable that a certain amount of the nuclear explosion products reached the lake, although U is found in elevated concentrations in fossil fuels, including coal [94]. Limited amounts of U are found in the minerals of apatite–nepheline deposits, for example, rinkite, monazite, lovchorrite, and uranium pyrochlore $(U,Ca,Ce)_2(Nb,Ti,Ta)2O_6(OH,F)$ [52]. Since the 1990s, there has been a

constant decrease in the U content, and its level in the surface sediment layers is comparable to the background (Figure 10).

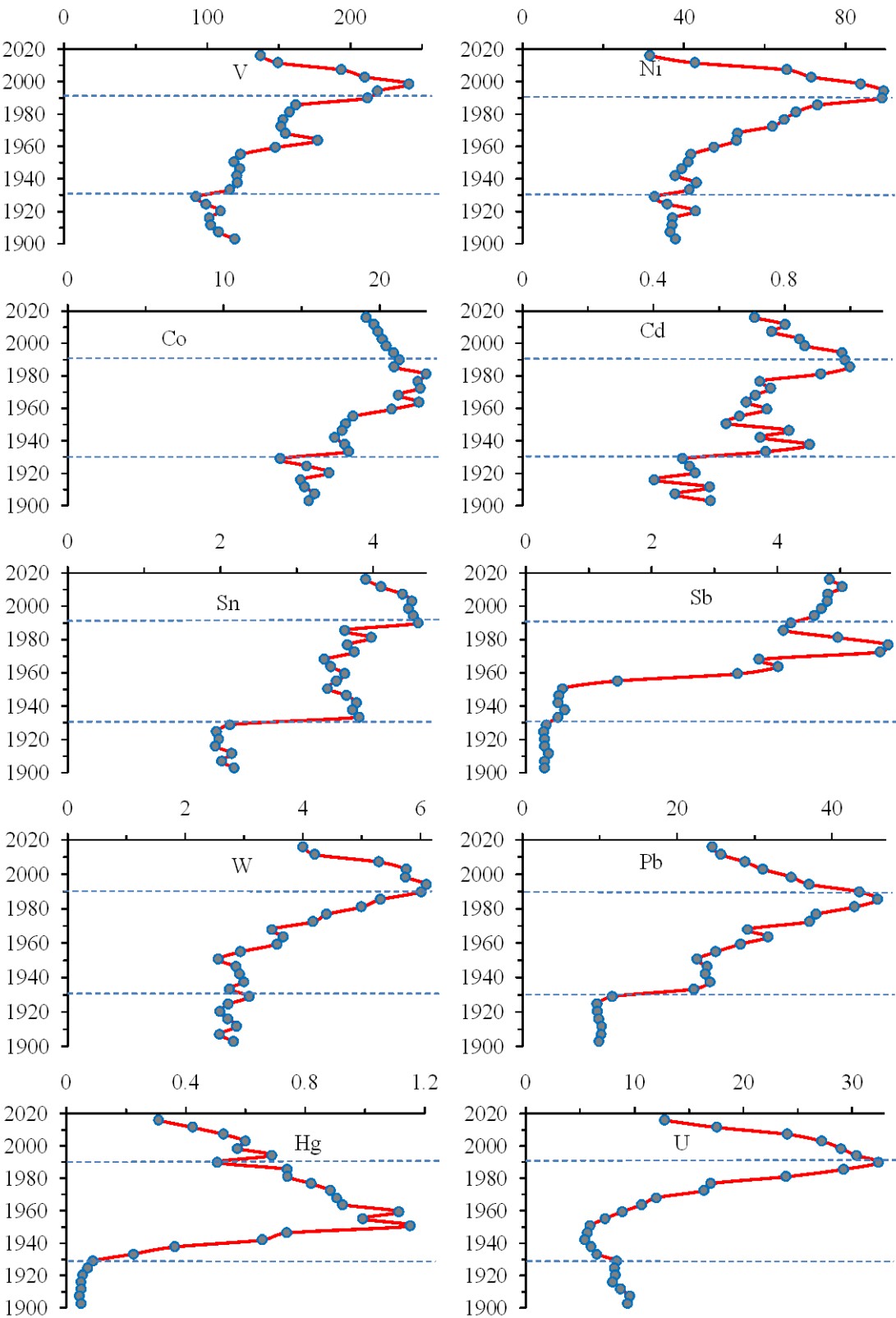

**Figure 10.** Vertical distribution of the content of elements (µg/g) that reduce the content in the upper sediment layers of Lake Bolshoy Vudjavr.

Aerotechnogenic transfer of substances also contributes to the pollution of Lake Bolshoy Vudjavr. These substances come from the plant processing of copper–nickel ores of Kola MMC, located in Monchegorsk, 45 km from the studied reservoir [95]. First of all, this applies to such elements as Cu, Ni, Co, and Zn. Moreover, the impact of emissions from Kola MMC on the environmental objects located hundreds of kilometers from the enterprise is wellknown [96]. The patterns of metal accumulation in the lake catchment area are different. For example, it was found that the input of Ni in the wastewater from the Severonickel Plant is eight times higher than the input from the catchment area (i.e., in the form of aerotechnogenic fallout), while the Cu input from the catchment area is 7.5 times higher than that from the wastewater [16]. This was reflected in the distribution of Cu and Ni in the sediment core of Lake Bolshoy Vudjavr, since the Cu concentration in the upper sediment layer exceeds the Ni concentration by a factor of seven, and Ni reduces the content after 1990, while Cu concentrations continue to increase during this period (Figures 10 and 11). A similar distribution pattern was noted for Co and Zn—if the first element decreases its content in the upper sediment layers, the latter has been stable for the last 30 years (Figures 10 and 11).

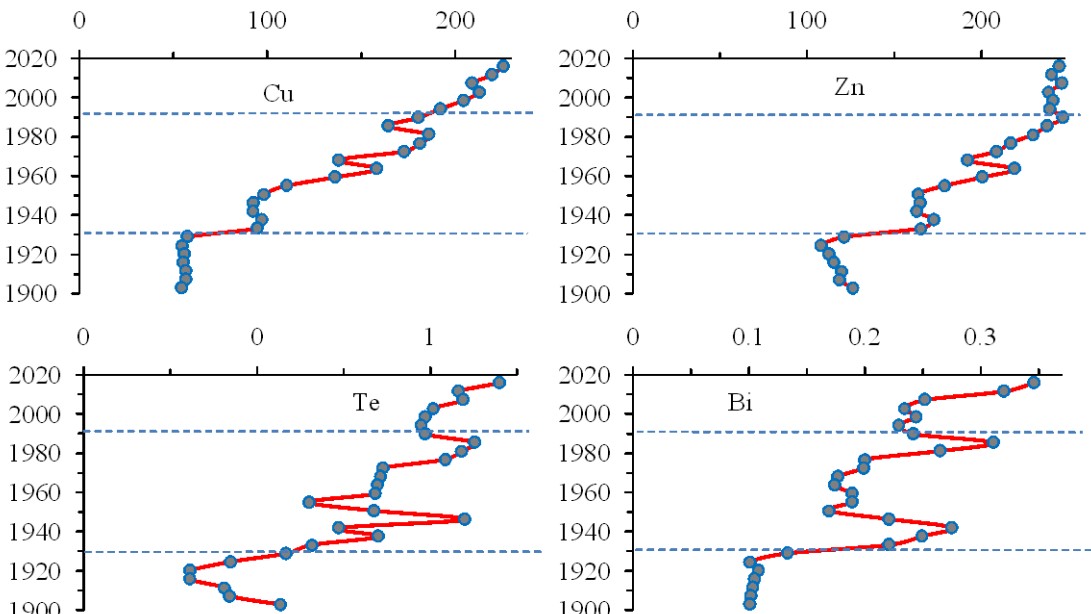

**Figure 11.** Vertical distribution of the content (μg/g) of Cu, Zn, Te, and Bi in the sediments of Lake Bolshoy Vudjavr.

Metals such as chalcophile Pb, Cd, and Hg classified as global pollutants [97], actively accumulate in lake sediments as a result of long-range atmospheric transport, which is confirmed by numerous studies in Murmansk region and neighboring regions [98–100]. At the same time, Pb from vehicle exhaust gases could also actively enter the ecosystem of Lake Bolshoy Vudjavr, since the studied reservoir has an urban status. From the 1930s to the early 2000s, tetraethyl lead is known to have been added to gasoline to improve fuel quality [101]. The decrease in Pb concentrations in the upper sediment layers of Lake Bolshoy Vudjavr is most likely due to the ban on the production of leaded gasoline in Europe and in Russia in particular. A similar pattern of Pb distribution can also be noted in the sediments of lakes in other cities of Murmansk region and the Republic of Karelia [88,93]. Pb, along with Sb, is also usually ranked among the priority pollutants of small lakes in the urbanized environment of the north of Russia.

The vertical distribution of Hg in the sediments of Lake Bolshoy Vudjavr clearly shows a surface maximum of up to 1.1 μg/g at a depth of 13–16 cm (Figure 10), which is more than 20 times higher than the average background concentrations of this extremely

toxic chalcophile element in the sediments of lakes in Murmansk region [21,98]. A similar distribution of Hg was found in the sediments of Belaya Guba of Lake Imandra [102]. A comparison with the vertical profiles of other pollutants leads to the conclusion that the increase in Hg contents in lakes Bolshoy Vudjavr and Imandra coincides with the onset of the wastewater inflow from "Apatite" JSC (P, Al, Ca, Na, Sr, K) and "Severonickel" (Ni, Cu, Co), but the maximum Hg concentrations were recorded earlier in time than those of the main polluting heavy metals (Ni and Cu). It can be assumed that the maximum influx of Hg occurred in the 1930s–1950s (Figure 10). During blasting operations at the first stages of development of apatite–nepheline deposits, fulminate mercury (mercury salt of fulmic acid $Hg(CNO)_2$) was used in blasting caps. During the Second World War, on the basis of the "Apatite" Plant, there was a workshop for manufacturing incendiary phosphorus bombs, in which the fulminate mercury was also used as an igniter.

Another chalcophile element in the sediments of Lake Bolshoy Vudjavr, tin, shows a distribution pattern similar to Pb and Cd (the correlation coefficient of Sn with these metals is 0.83 and 0.88, respectively)—a rather sharp increase in the early 1930s, and a decrease after the 1990s (Figure 10). At the same time, high values of the correlation coefficient Sn were recorded with all alkali (except Li) and alkaline earth metals and REE (r > 0.9). Sn possibly enters the lake sediments both as a part of mine and concentrating plant effluents, and as part of emissions from fossil fuel combustion and copper–nickel production. The latter conclusion is supported by a more than 50-fold increase in the Sn content in the sediments of Lake Rybachy, located 50 km from Pechenganickel plant, and a 15-fold increase in Sn in the sediments of Lake Severnoye compared to the background concentrations (Table 2) [27,83].

### 3.8. Trace Elements That Increasethe Content in the Upper Sediment Layers

Minerals containing niobium are widely distributed in the rocks of the Khibiny alkaline massif. Niobium is always accompanied by tantalum, and the content of both metals is very closely correlated (r = 0.98) in the sediments of Lake Bolshoy Vudjavr. The close chemical properties of Nb and Ta determine their joint presence in the same minerals and participation in common geological processes. A number of titanium-containing minerals: titanite (sphene) $CaTi[SiO_4]O$, perovskite $CaTiO_3$, ilmenite $FeTiO_3$, ilmenorutil $(Ti,Nb,Fe)_3O_6$, rutile $TiO_2$, and rinkite $Na_2Ca_4(Ce,Y)Ti[Si_2O_7]_2(F,OH,O)_4$, are minerals common in the Khibiny [52]. Niobium replaces Ti in titanium-containing minerals, as evidenced by the very close correlation of Ti with Nb and Ta (r = 0.97). The form of finding niobium can be different: scattered (in rock-forming and accessory minerals of igneous rocks) and mineral. In total, more than a hundred minerals containing niobium are known, and many of them are found in the Khibiny massif. Of these, the following minerals are of industrial importance in the Khibiny: pyrochlore $(Na, Ca)_2Nb_2O_6(OH, F)$ ($Nb_2O_5$ up to 62%), loparite $(Ce,Na,Ca)_2(Ti,Nb)_2O_6)$ ($Nb_2O_5$ up to 20%), lovchorrite $(Na(Ca, Na)_2(Ca, Ce)_4TiO_2F_2(Si_2O_7)_2)$ ($Nb_2O_5$ up to 2%, $ThO_2$ up to 0.5–1%, and rare earth oxides $TR_2O_3$ up to 17%) [52]. The concentrations of Nb and Ta in the surface and background layers of the sediments of Lake Bolshoy Vudjavr are 1–2 orders of magnitude higher than the content of these elements in lakes outside the Khibiny (Table 2, Figure 12), which indicates the natural enrichment of this mountain massif with these metals.

Fifty thousand tons of sphene concentrate at the "Apatit" Plant were planned to be produced during the five-year plan of 1933–1937 [14]. Between 1934 and 1939, a lovchorrite mine was developed, and during this time, 19.4 thousand tons of lovchorrite ore were mined [86]. The content of Nb and Ta in the sediments of Lake Bolshoy Vudjavr, as well as Ti, increased sharply in the early 1930s since the beginning of the development of apatite–nepheline deposits and then only increased.

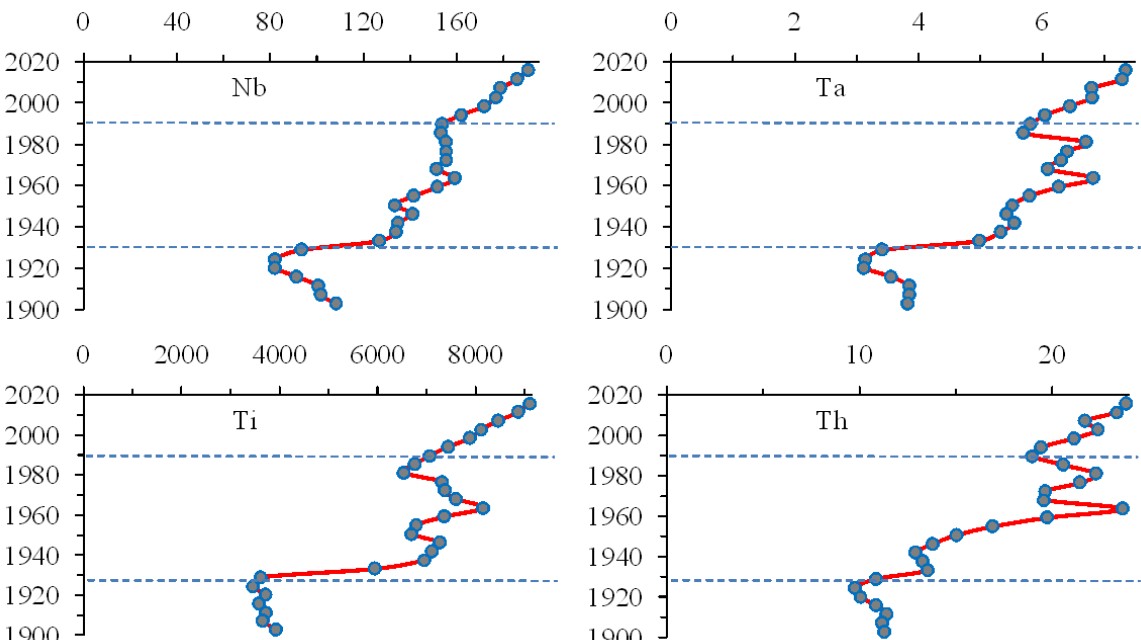

**Figure 12.** Vertical distribution of the content (µg/g) of Nb, Ta, Ti, and Th in the sediments of Lake Bolshoy Vudjavr.

Thorium is present in small amounts (from fractions to a few %) in almost all minerals containing Nb and Ta (lovchorrite, rinkite, loparite, pyrochlore). Thorium is almost always present in the minerals of REE, which serve as one of the sources of its production (for example, in monazite Ce[PO$_4$]). Therefore, Th closely correlates (r = 0.94) with Nb and Ta in the sediments of Lake Bolshoy Vudjavr, as well as with Ti (r = 0.87). All these four elements have similar distribution pattern (Figure 12).

Bismuth and tellurium show a very similar distribution pattern in the sediments of Lake Bolshoy Vudjavr, resembling that of other chalcophile elements (Figure 11), and their sources are probably emissions from fossil fuel combustion and copper–nickel production. The latter conclusion is supported by a more than 17-fold increase in the Bi content in the sediments of Lake Rybachy and a 7-fold increase in the sediments of Lake Severnoe compared to the background concentrations (Table 2) [27,83]. Since Bi and Te are not included in any of the described minerals of the Khibiny massif [52], these elements are most unlikely to have come into the lake with the effluents of the mines and the processing plant.

### 3.9. Trace Element Fractions in Sediments

Analysis of the forms of elements in the sediments of Lake Bolshoy Vudjavr showed that the studied elements are mainly found in stable compounds—mineral, acid-soluble, and organic. For example, Ni, Cu, Zn (Figure 13), Cd, W, V, Pb, Sb, and Sn are predominantly found in mineral phases throughout the entire depth of the studied sediment core, that is, in the most forms difficult to be involved in biogeochemical cycles. At the same time, the predominant phase for chalcophile elements Ni, Cu, Zn, Cd, Pb, W, and V, in addition to mineral forms, is the fraction associated with organic matter. Copper exhibits the most pronounced organophilic properties throughout the sediment core of Lake Bolshoy Vudjavr (Figure 13).

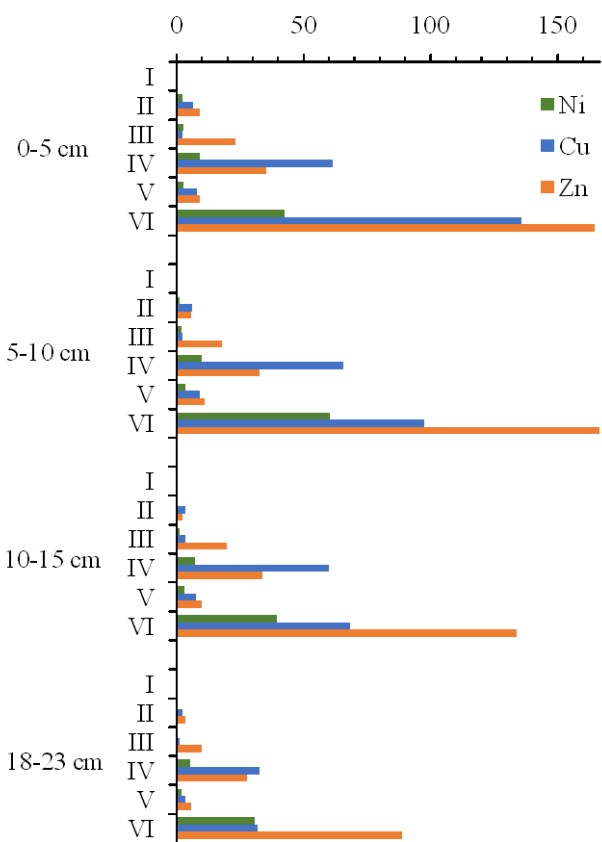

**Figure 13.** Concentrations (µg/g) of various forms of Ni, Cu, and Zn in the sediments (layers 0–5, 5–10, 10–15, and 18–23 cm) of Lake Bolshoy Vudjavr, forms: I—water-soluble forms, II—exchangeable cations (mobile forms), III—forms associated with Fe and Mn hydroxides, IV—forms associated with organic matter, V—acid-soluble forms, and VI—mineral forms.

Among all the metals studied in this work, the largest relative amount of mobile and most potentially bioavailable (the sum of one and two fractions) forms over the entire depth of the core was found for Cd. In contrast to Cd, Sn is the most strongly bound metal, for which the absolute predominance of mineral forms was revealed throughout the core. For individual metals, Zn and Pb among them, the bond with oxides and hydroxides of Fe and Mn is essential throughout the entire depth of the investigated lake sediment core. In many respects, the established patterns are close to those obtained for the same elements in the sediments of Lake Komsomolskoye (Monchegorsk, Murmansk region) and in the sediments of small lakes in Petrozavodsk and Murmansk [71,77,103].

*3.10. Anthropogenic Load on the Lake*

The contamination of the sediments of Lake Bolshoy Vudjavr was assessed by the integral indicator PLI (Pollution Load Index) and the contamination factor (CF), where all the analyzed elements were taken into account, and the average concentrations of elements in the sediment layers of 24–27 cm, accumulated before the development of apatite–nepheline deposits started, were used as the background level. The value of the PLI increases from the lower layers to the upper ones (Figure 14). The contamination of the sediment layers of Lake Bolshoy Vudjavr, formed before the onset of active anthropogenic impact on the reservoir (20–27 cm), can be assessed as low. After the start of the activity of the "Apatite" Plant and the release of a large amount of runoff from the mines into the lake, a sharp increase in the PLI occurred and the sediment layers accumulated from the 1930s to the mid-1950s (14–20 cm), characterized by a moderate level of pollution. From the mid-1950s to the late 1980s, there was a gradual increase in the PLI values associated with the commissioning of new mines and an increase in the capacity for mining and processing

apatite–nepheline ore, as well as a sharp increase in using automobile and fossil fuels (coal and masut), containing a large amount of heavy metals (Pb, Ni, V, W, see Figure 10). The highest PLI values (on the border between moderate and high pollution) were detected in the layers from 2 to 7 cm, which were accumulated from the early 1990s to the mid-2000s. The maximum concentrations of most of the elements present in the apatite–nepheline ore, including P, Ti, Nb, Ta, REE, alkali and alkaline earth metals, as well as heavy metals, are found precisely in these layers of the studied sediments (Figures 5–7, 11 and 12). Since the beginning of the 2000s (from 4 cm and above), the level of pollution has been gradually decreasing in the most recent sediment layers of Lake Bolshoy Vudjavr, which correlates well with a decrease in the content of individual elements, such as REE and heavy metals, in the surface sediment layers. Nevertheless, in general, the PLI values in the uppermost sediment layers remain on the border between moderate and high pollution (Figure 14).

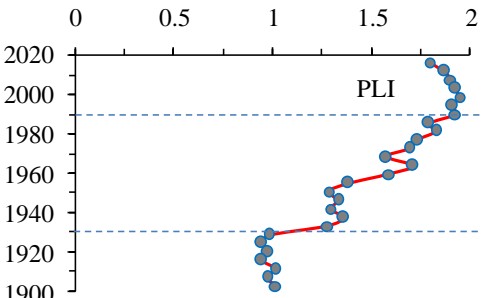

**Figure 14.** Distribution of Pollution Load Index (PLI) value in the sediment core of Lake Bolshoy Vudjavr.

Antimony (18.2), P (10.3), and Sr (7.8) have the maximum values of the contamination factor CF (high according to the classification [64]), as well as La (6.0) from the REE. The values of the contamination factor CF for the rest of the REE as well as Pb (4.6), Cu (4.0), U (3.5), and Bi (3.4) are also considerable.

## 4. Conclusions

Being the largest inland reservoir of the Khibiny alkaline mountain range, Lake Bolshoy Vudjavr has been experiencing intense anthropogenic pressure for more than 90 years, since the development of the richest apatite–nepheline deposits in the world started in 1929. A significant volume of raw materials containing ore minerals is currently stored in the tailings of apatite–nepheline factories. The development of apatite–nepheline deposits of the Khibiny alkaline massif led to the formation of runoff and emissions of the main rock-forming and trace elements into the environment and their subsequent sedimentation in the previously untouched mountain environment.

As a result of studying the sediments of Lake Bolshoy Vudjavr, elevated concentrations of alkali and alkaline earth metals, rare earth elements (REE), P, many trace elements, including heavy metals (Pb, Sn, Sb, Cu), associated with the extraction and processing of apatite–nepheline ores and the combustion of fossil fuels, were found in the surface sediment layers. Along with the macroelements extracted from apatite–nepheline ores at the processing plants at the present time, the increased content of Zr, Hf, and Mo was detected in the sediments of Lake Bolshoy Vudjavr in the 1930s, which is associated with the exploration and development of Zr raw materials and Mo ores at that time. The content of these elements decreased further to the surface, as the development of the raw materials was terminated. The content of Nb, Ta, Ti and Th increased sharply in the early 1930s due to the mining of sphene, loparite, and lovchorrite ores. Unlike the first three elements (Zr, Hf, and Mo), the content of the latter four in the lake sediments continues to increase to the surface layers, despite the cessation of mining of the loparite and lovchorrite ores.

Analysis of the forms of the studied elements in the sediments of Lake Bolshoy Vudjavr showed that they are mainly found in hard-to-reach (stable) fractions—mineral,

acid-soluble, and organic, which indicates that they came into the lake sediments mainly as part of the runoff from apatite–nepheline mines and atmospheric fallout after fuel combustion. For example, Ni, Cu, Zn, Cd, W, V, Pb, Sb, and Sn are predominantly found in mineral phases throughout the entire studied sediment core, i.e., in the most difficult forms to be involved in biogeochemical cycles. At the same time, for Cd, Cu, Pb, Zn, Ni, W, and V, in addition to mineral forms, the predominant phase is the fraction associated with organic matter.Therefore, the latter elements can have a negative impact on hydrobionts, primarily on benthic organisms.

An assessment of the degree of pollution of the sediments of Lake Bolshoy Vudjavr using the integrated Pollution Load Index (PLI) revealed a high level of pollution. A sharp increase in the PLI value occurs first in the activity of the mining enterprise from the 1930s. This pollution reached its maximum in the late 1980s during the period of highest extraction of apatite–nepheline ores and has yet to decrease.

The diversity of the mineral composition of the apatite–nepheline deposits of the Khibiny and the mining activity predetermined the entry into Lake Bolshoy Vudjavr of a large list of elements that from the beginning of the deposits development to the present violate the natural geochemical cycles of elements and alter the chemical composition of the water column and lake sediments. As a result, the habitat of the flora and fauna of the mountain lake is changing, which, due to the natural environment, is vulnerable and slowly restoring natural resources.

Violation of the natural geochemical cycles of macroelements and rare elements, including alkaline, alkaline earth, and rare earth elements, causes serious changes in biological systems of various levels of their organization: from the organism to the ecosystem. The disintegration of rocks during technological processes increases the mobility of these elements, and increases their concentration in water and sediments. These elements are able to accumulate in the lake ecosystem, enter the food chain, and have a prolonged negative impact on living organisms, including the population health. Nevertheless, many elements accumulating in sediments (both macro- and microelements) are not given due attention in environmental practice. Many of these elements are not included in the lists of controlled elements; their behavior in natural environments and their impact on living organisms has not been studied. Therefore, a large number of uncontrolled elements can not only have a synergistic suppressive effect on lake hydrobionts, but also pose a serious threat to the health of the city's population. Lake Bolshoy Vudjavr is a mining wastewater collector, and at the same time, the lake is an additional source of drinking water supply for Kirovsk. The problem of pollution of water supply sources in settlements is relevant for the population of the Murmansk region due to the presence of a large number of mining enterprises.

One of the solutions to the problem of intense pollution of Lake Bolshoy Vudjavr and other water bodies polluted by mining effluents can be the integrated development of the subsoil, which is one of the components of the theory of sustainable development. A large number of components that are valuable and necessary for the country's economy can be extracted from the rocks of the apatite–nepheline deposits of the Khibiny mountain alkaline massif (raw materials for the extraction of Ti, Zr, Mo, Sr, F, Nb, Ta, Th, REE, etc., in addition to the currently extracted apatite and nepheline). The integrated use of mineral raw materials will reduce the load on the environment due to a decrease in the volume of industrial waste.

**Author Contributions:** Conceptualization, V.D., Z.S. and D.D.; methodology, Z.S., V.D. and A.G.; software, D.D. and Z.S.; validation, V.D., Z.S. and D.D.; investigation and resources, V.D., Z.S., D.D. and A.G.; data curation, V.D. and Z.S.; writing—original draft preparation, V.D. and Z.S.; writing-review and editing, V.D. and Z.S.; visualization, V.D., Z.S. and D.D., supervision, V.D.; project administration and funding acquisition, V.D. and Z.S. All authors have read and agreed to the published version of the manuscript.

**Funding:** The research is supported by the Russian Science Foundation (project No. 22-27-00131).

**Acknowledgments:** The authors sincerely thank the colleagues from Petrozavodsk and Apatity: P.M. Terentjev for his help in lake sediment sampling, as well as to our colleagues O.P. Korytnaya, A.S. Paramonov, S.V. Burdyukh, M.V. Ekhova, V.L. Utitsina, and A. Fedorov for conducting high-quality analytical studies.

**Conflicts of Interest:** The authors declare no conflict of interest.

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
