# Peer review of "A Paleolimnological Perspective on Arctic Mountain Lake Pollution"

_water, doi:10.3390/w14244044_

Round 1

Reviewer 1 Report

To reconstruct the history of pollution and assess the intensity of anthropogenic impact on the largest Khibiny reservoir, this study analyzed the input elements into the lake located in close proximity to the mines. This manuscript is well written with good technical quality, concise experimental scheme, methods, and data. However, this paper is a very localized paper that is confined to Russia, hence, I think the authors could provide more general environment protection proposals for mountain lakes (or specifically for Arctic Mountain lakes) based on the findings of this study. Since this manuscript was submitted to the journal of Water, I think the effect of the water (runoff) or regional climate on the pollution migration from the mining area to the lake could be analyzed or discussed. Additionally, there are some mistakes in this manuscript (especially in tables and figures). The authors should carefully proofread the entire manuscript.

Author Response

Response to Reviewer 1 Comments

Dear Unknown Reviewer,

Thank you very much for your attentive and sensitive attitude to our manuscript.

We have rewritten the manuscript according to Your comments. We are grateful to reviewer for useful comments that will certainly improve the quality of our manuscript.

Below are our responses to the reviewer's comments highlighted in red.

Point 1: However, this paper is a very localized paper that is confined to Russia, hence, I think the authors could provide more general environment protection proposals for mountain lakes (or specifically for Arctic Mountain lakes) based on the findings of this study. Since this manuscript was submitted to the journal of Water, I think the effect of the water (runoff) or regional climate on the pollution migration from the mining area to the lake could be analyzed or discussed.

Response 1: The comments of the reviewer are very useful and the answers to them are in the conclusion. The conclusion was rewritten taking into account the comments of the reviewers.

Point 2: Additionally, there are some mistakes in this manuscript (especially in tables and figures). The authors should carefully proofread the entire manuscript.

Response 2: The authors carefully reviewed the entire manuscript, including Tables and Figures, and corrected the mistakes.

Thank you very much again for Your help in improving the quality of our manuscript.

Sincerely Yours

Vladimir

Prof. Dr. Sci. Vladimir Dauvalter

Leading Scientific Researcher

Laboratory of Freshwater Ecosystems

Institute of North Industrial Ecology Problems

Kola Science Centre, Russian Academy of Sciences

14a Akademgorodok, INEP, Apatity,

Murmansk region, 184209, Russia

Phone: +7 81555 79774

E-mail: vladimir@dauvalter.com

Reviewer 2 Report

The manuscript reports a well-developed and adequately described study. All necessary data are adequately elaborated. 

The main problems in the manuscript concern discussions and conclusions. The discussions should be separated from the results. Conclusions should be rewritten taking into account the purposes set in the introduction. One important issue has not been considered: Based on the results, Are the studied sediments polluted?

The manuscript  needs deep revision by an English-speaking expert.

Some comments to improve the quality of the manuscript are given directly on the pdf version of the manuscript

Author Response

Response to Reviewer 2 Comments

Dear Unknown Reviewer,

Thank you very much for your attentive and sensitive attitude to our manuscript.

We have rewritten the manuscript according to Your comments. We are grateful to reviewer for useful comments that will certainly improve the quality of our manuscript.

Below are our responses to the reviewer's comments highlighted in red.

Point 1: Title too long.

Response 1: The title of the article has been shortened – “A Paleolimnological Perspective on Arctic Mountain Lake Pollution”.

Point 2: Abstract, lines 18-19 – Unclear sentence, to be rewritten: “The elements in the sediments are mainly found in hard-to-reach fractions – mineral, acid-soluble and organic.”

Response 2: The sentence was rewritten: “Analysing the forms of elements in the lake sediments showed that the studied elements are mainly found in stable fractions – mineral, acid-soluble and associated with organic matter.”

Point 3: Introduction, line 41 – add references.

Response 3: The reference was added.

Point 4: Introduction, line 43 – add references.

Response 4: The reference was added.

Point 5: Introduction, line 44 – add references.

Response 5: The reference was added.

Point 5: Introduction, line 46 – add references.

Response 5: The reference was added.

Point 6: Introduction, line 48 – add more than one reference.

Response 6: The references were added.

Point 7: Lines 48-49 – Unclear sentence, to be rewritten: “The concentrations of elements elevated relative to the natural background content 48 negatively affect the water quality in the lakes surrounding regional mining centers.”

Response 7: The sentence was rewritten: “The input of elements in elevated relative to natural background contents adversely affects the quality of water in lakes that surround concentrated mining centers.”

Point 8: Introduction, line 51 – add more than one reference.

Response 8: The references were added.

Point 9: Introduction, line 52 – add more than one reference.

Response 9: The references were added.

Point 10: Introduction, line 53 – add references.

Response 10: The references were added.

Point 11: Introduction, line 53 – What do you mean: “technogenic impact” ?

Response 11: “technogenic impact” was chanched “anthroponogenic impact”.

Point 12: Introduction, line 52 – add more than one reference.

Response 12: The references were added.

Point 13: Introduction, line 56 – add references for each concept expressed in the sentence.

Response 13: The references were added.

Point 14: Introduction, line 61 – What do you mean: “temporal evolution”?

Response 14: The sentence “However, to date, only a limited number of studies focused on the temporal evolution of element accumulation in the lakes of the Russian Arctic and the Khibiny Mountains in particular” was chanched “However, to date, only a limited number of studies have been devoted to the change in the elements accumulation in recent centuries in the lakes of the Russian Arctic and the Khibiny Mountains in particular”.

Point 15: Introduction, lines 65, 67 and 69 – add references.

Response 15: The references were added.

Point 16: Introduction, line 70 – “Trace elements” such as?

Response 16: it was added “Trace elements, such as Pb, Sr, rare earth elements (REE)”

Point 17: Introduction, lines 71 and 72 – add references.

Response 17: The references were added.

Point 18: Introduction, lines 73 and 74 – add references for each concept expressed in the sentence.

Response 18: The references were added.

Point 19: Lines 73–75 – Unclear sentence, to be rewritten: “with high resolution and a large list of analyzed elements specifically focusing on recent industrial growth and its ecological heritage’.

Response 19: The sentence was rewritten: “with the determination of the sedimentation rate and a large list of analyzed elements.”

Point 20: Line 77 – “This study of sediments in Lake Bolshoy Vudjavr pays special attention to the industrial era.”

Response 20: The sentence “This study of sediments in Lake Bolshoy Vudjavr pays special attention to the industrial era” was chanched “This study of sediments in Lake Bolshoy Vudjavr focuses on the industrial era”.

Point 21: Line 79 – Did you study Lake Bolshoy Vudjavr or Khibiny resevoir? : “Khibiny reservoir”.

Response 21: The words were rewritten: “Khibiny lake.”

Point 22: Line 79 – Which ones of elements?

Response 22: The words were rewritten: “a large list of elements.”

Point 23: Line 81 – “The studied sediment core was contaminated by major rock-forming and trace elements” Can the major elements be contaminants? If they are major by definition, they cannot be contaminants.

Response 23: The sentence was rewritten: “The studied sediments accumulate major rock-forming and trace elements.”

Point 24: Line 87 – It is appropriate to implement Figure 1 so as to have a view of the study area within Russia.

Response 24: Figure 1 modified according to reviewer's comment.

Point 25: Line 88 – Insert link to Figure 1.

Response 25: Link to Figure 1 has been inserted.

Point 26: Line 133 – add references.

Response 26: The references were added.

Point 27: Lines 138–141 – Show them (mines) on the map.

Response 27: The sentence: “The following mines are currently operating: Kirovsky (Kukisvumchorr and Yukspor deposits), Rasvumchorrsky (Apatite Circus and Rasvumchorr Plateau deposits), Vostochny (Koashva and Nyorkpakhk deposits) and the recently discovered Oleniy Ruchey (Koashva deposit). “ was rewritten: “The following mines are currently operating: in the southeastern part of the Khibiny massif (catchment of the largest lake in the Murmansk region - Imandra) – Kirovsky (Kukisvumchorr and Yukspor deposits) and Rasvumchorrsky (Apatite Circus and Rasvumchorr Plateau deposits) (Figure 1), in the southwestern part of the Khibiny massif (catchment of the deepest lake in the Murmansk region - Umbozero) – Vostochny (Koashva and Nyorkpakhk deposits) and the recently discovered Oleniy Ruchey (Koashva deposit) (outside of Figure 1).“

Point 28: Lines 147–149 – Can you add a reference regarding this event? “a public environmental movement was formed advocating assigning the status of national park to the Khibiny and prohibiting their further development“.

Response 28: The reference was added.

Point 29: Line 224 – 2.5.1. Sediment sampling – Explain in detail the methodologies used to take the samples. Instruments, weight and volume of samples and describe all sampling steps.

Response 29: The section has been expanded to describe the details of sediment sampling.

Point 30: Line 239 – 2.5.3. Sediment dating – Explain in detail the methodologies and describe all steps.

Response 30: The section has been expanded to describe the details of sediment datling.

Point 31: Line 273 – 2.5.6. Determining the chemical fractions of elements – Explain in detail the methodologies and describe all steps.

Response 31: The section has been expanded to describe the details of sediment datling.

Point 32: Line 303 – discussions should be separated from results 3. Results and Discussion

Response 32: The logic of the article is constructed in such way that it is impossible to separate the presentation of the research results and the discussion. We ask the reviewer and the editors of the journal to leave this part in a combined form.

Point 33: Line 345 – this chapter (3.3. The Main Rock-Forming Elements) should be shortened.

Response 33: A significant increase in the content of a large list of the main elements in the sediments of Lake Bolshoi Vudjavr was recorded, associated with the influence of apatite-nepheline production. Much attention is paid to the accumulation of these elements in the article, so the reduction of this section of the article is considered undesirable.

Point 34: Line 513 – this chapter (3.6. Trace Elements that Reduce the Content throughout the Sediment Core) should be shortened.

Response 34: The description of these elements is given in only three short paragraphs, and the authors consider it impossible to shorten this section.

Point 35: Line 551 – this chapter (3.7. Trace Elements that Reduce the Content in the Upper Sediment Layers) should be shortened.

Response 35: A large list of elements, the supply of which is mainly associated with atmospheric pollution of the lake, is described in this section. These elements are the most important and dangerous pollutants of the lake, therefore, quite a lot of attention is devoted to their description. Please allow not to reduce the size of this section due to the importance of their definition for assessing the degree of pollution of the lake.

Point 36: Line 761 – Lack of "discussion" section makes it difficult to read the paper and understand the conclusions.

Response 36: The conclusion of the article has been rewritten in accordance with the comments of the reviewers. Regarding the separate description of the sections "Results" and "Discussion", the explanation is given above.

Point 37: Lines 765–768 – Is it useful?

Response 37: Two sentences were deleted from the conclusion.

Point 38: Lines 774–776 – Is it useful?

Response 38: The sentence was deleted from the conclusion.

Point 38: Lines 800–817 – Is it useful? Why don't you discuss the negative impact mining has on the quality of the environment? Possible toxic effects on humans, animals and vegetation?

Response 38: The last paragraph has been deleted from the conclusion. The conclusion was rewritten taking into account the comments of the reviewers.

Thank you very much again for Your help in improving the quality of our manuscript.

Sincerely Yours

Vladimir

Prof. Dr. Sci. Vladimir Dauvalter

Leading Scientific Researcher

Laboratory of Freshwater Ecosystems

Institute of North Industrial Ecology Problems

Kola Science Centre, Russian Academy of Sciences

14a Akademgorodok, INEP, Apatity,

Murmansk region, 184209, Russia

Phone: +7 81555 79774

E-mail: vladimir@dauvalter.com
